# *Jamais Vu*: Exposing the Generalization Gap in Supervised Semantic Correspondence

Octave Mariotti[1]    Zhipeng Du[1]    Yash Bhalgat[2]    Oisin Mac Aodha[1]    Hakan Bilen[1]

[1]University of Edinburgh    [2]University of Oxford

https://github.com/VICO-UoE/JamaisVu

## Abstract

The goal of semantic correspondence (SC) estimation is to establish semantically meaningful matches across different instances of an object category. In this work, we illustrate how recent supervised SC methods generalize poorly beyond the annotated keypoints seen during training, thus effectively acting as keypoint detectors. To address this, we propose a new approach for learning dense correspondences by lifting 2D keypoints into a canonical 3D space using monocular depth estimation. Our method constructs a continuous canonical manifold that captures object geometry without requiring explicit 3D supervision or camera annotations. Additionally, we introduce SPair-U, an extension of SPair-71k with novel keypoint annotations, to better assess generalization. Experiments not only demonstrate that our model significantly outperforms supervised baselines on unseen keypoints, highlighting its effectiveness in learning robust correspondences, but that unsupervised baselines outperform supervised counterparts when evaluated across different datasets.

## 1    Introduction

Semantic correspondence (SC) estimation involves identifying semantically matching regions in images across different instances of the same object category. It remains a challenging problem, as it requires recovering fine-grained details while maintaining robustness against variations in object appearance, shape, and viewing conditions. Recent advances in large-scale vision models, particularly self-supervised transformers [6, 39] and generative diffusion models [41], have led to notable improvements in SC. When employed as backbones, these models have achieved over 20% gains in accuracy on the SPair-71k benchmark [34]. However, despite these advances, recent studies have highlighted that these powerful representations often struggle to disambiguate symmetric object parts due to their visual similarity [62, 32].

SC methods can be broadly categorized into two groups in terms of supervision: unsupervised models, which do not require correspondence annotations during training [1, 2, 61, 32], and supervised models [8, 18, 62], which are trained on manually annotated correspondences. As expected, supervised models generally achieve higher performances when using the same backbone and same training set as unsupervised models. However, a key limitation of current benchmarks is that evaluation is typically performed on the *same* set of keypoints used for training, potentially inflating perceived generalization. As illustrated in Fig. 1, the performance of supervised models drops significantly when evaluated on unseen keypoints, while unsupervised models maintain their performance.

Building dense correspondences are key to fine-grained object understanding and improving robustness in various recognition tasks, and essential to many applications including texture transfer [38] and robotic manipulation [49]. In this work, we examine the performance of state-of-the-art SC

39th Conference on Neural Information Processing Systems (NeurIPS 2025).

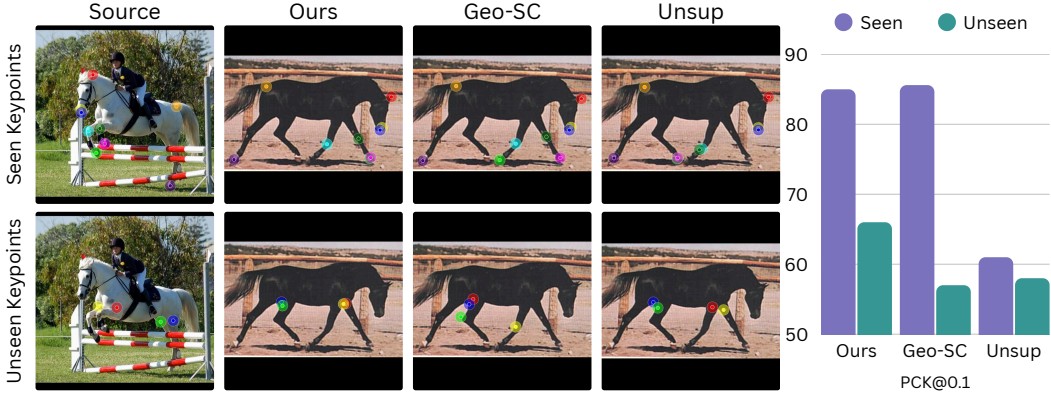

Figure 1: **Illustration of the generalization gap on unseen keypoints.** (Left) Top row: when evaluated on known keypoints, both our model and Geo-SC [62] perform well, while the unsupervised DINOv2+SD [61] struggles to correctly disambiguate the legs of the horse. Bottom row: when presented with keypoints unseen at training time, both our model and DINO+SD predict noisy but reasonable correspondence, while Geo-SC predictions noticeably degrade. (Right) Even though it obtains strong performance on known keypoints, Geo-SC performs worse than its unsupervised counterpart on our new benchmark of unseen keypoints. In comparison, our model still achieves competitive results.

models when evaluated on points that lie outside the set of annotated training keypoints. Under these conditions, we observe that supervised pipelines often underperform their unsupervised counterparts, effectively reducing their function to that of 'sparse keypoint detectors'. We attribute this limitation to two main factors: (i) the sparsity of supervision, which typically focuses on a limited number of keypoints and (ii) the lack of evaluation on unseen points, which tends to favor models that bias their predictions toward the nearest seen annotations.

We argue that an ideal SC method should be capable of matching arbitrary points, akin to the objectives in classical dense correspondence tasks [26]. To move beyond sparse supervision, we propose a learning framework that predicts dense feature maps and supervises them using geometry from an off-the-shelf depth estimator, thereby enabling training on richer and more spatially diverse cues. Some unsupervised SC methods [46, 32] have leveraged 3D geometry to learn dense correspondences through mapping object pixels to a spherical coordinate system where each coordinate corresponds to a different characteristic point of the object. However, this approach requires estimating the object shape and viewpoint from a collection of 2D images, which limits the applicability of such models to synthetically generated datasets [46] and they can require additional camera viewpoint supervision [32].

We propose a new approach that leverages existing 2D keypoint annotations and estimated 3D geometry to learn dense correspondences. We build on the idea of learning a canonical representation of the object category, which is invariant to the object instance, viewpoint, and pose. We achieve this by lifting the 2D keypoints to 3D using a monocular depth model, aligning them with a set of canonical keypoints that are shared across all instances of the object category. Finally, by interpolating between them, we learn a continuous canonical manifold, that captures the underlying 3D shape of the object and incorporates geometric constraints into learning more effective and general feature representations. We also introduce a new dataset for SC estimation, SPair-U, which extends the original SPair-71k test annotations with a set of new keypoints, allowing us to evaluate the generalization of SC models on unseen keypoints. We show that supervised SC models trained on the original SPair-71k dataset typically fail to generalize well to unseen keypoints, while our method is able to learn a more general representation that can be applied to unseen keypoints.

## 2 Related Work

**Supervised methods.** Supervised approaches rely on the availability of datasets with annotated keypoints such as CUB [51], PF-PASCAL [15], and SPair-71k [34] to learn corresponding points across instances of the same object class depicted in different images. This is typically using contrastive objectives minimizing distance between features coming from the same keypoints while

pushing other features away [16, 44, 61, 62]. A more computationally intensive option is to compute dense 4D correlations maps between each source and target locations [8, 33, 18, 22]. To obtain stronger descriptors, it is also common to aggregate features from multiple network layers to form hypercolumns [35, 1, 62]. Current state-of-the-art supervised methods forego training from scratch and instead typically use a large pretrained vision model as a backbone, the most popular options being DINOv2 [39] and Stable Diffusion [41]. While effective at matching instances of keypoints of the same type that have been observed during training, in our experiments we demonstrate that current supervised methods have a tendency to overfit to the set of keypoints observed during training and struggle to generalize to previously unseen keypoints (see Fig. 1 for an example).

There have also been recent attempts to utilize the expressive power of the representations encoded in large multi-modal models for detecting sets of keypoints. Few-shot methods require supervision in the form of a support set at inference time [28, 29, 17]. Zero-shot methods forego the need for such supervision, but instead require that keypoints should be described via natural language prompts [63, 60]. Describing common keypoints (e.g., 'the left eye') can be easily done via language, but how to best describe less salient points via text is not so clear. There have also been attempts to develop models that can take different various modalities (i.e., text and or keypoints) as input [30]. While promising, these methods make use of large multi-modal models and need large quantities of keypoint supervision data, spanning many diverse keypoints and categories, for pretraining.

**Unsupervised / weakly-supervised methods.** Methods that do not use correspondence supervision during training range from unsupervised approaches using general-purpose backbones [1, 61], very weakly supervised methods that only assume an curated training set without labels, e.g., images of a single category [46, 45, 2], zero-shot methods that only use test-time information about the relationships between keypoints [38, 62], dense methods that directly impose structure on the correspondence field [7], and weakly supervised methods that use extra labels like segmentation masks or camera pose [20, 32, 4]. Earlier unsupervised methods typically use self-supervised objectives that make use of synthetic deformations/augmentations of the same image [46] or by using cycle consistencies [45, 47, 48] to provide pseudo ground truth correspondence. Later it was observed that large pretrained vision models naturally posses features that are very strong for SC, despite not being trained on this task explicitly. As a result, more recent unsupervised methods tend to not train their own backbones from scratch and instead explore ways to aggregate [1, 61], or align [14, 32] these features across images.

**Geometry-aware methods.** Inspired by the classic correspondence setting in vision that relies on geometric constraints to match the same 3D locations across views [10, 42], utilizing geometry cues is an effective way to learn SC. For SC, the underlying assumption is that different instances from the same object category share a similar spatial structure. Flow and rigidity constraints are often used in tracking [53, 58] and unsupervised SC [25, 14, 4]. Recent studies have shown that ambiguities caused by symmetric objects are a major source of errors in SC [62, 32]. One potential way to mitigate these errors is to develop 3D-aware methods. Initially, this has been explored by building correspondence across images by matching points along the surface of objects to 3D meshes [64, 24, 37, 56, 11], but the requirement for meshes greatly limits the applicability of these approaches. More recently, methods have been proposed to learn 3D shape from image collections [36, 57, 3]. However, these methods require solving multiple problems at once, i.e., estimating object shape, deformations, camera pose, and are therefore limited to specific types of object categories and tend to break easily when applied to more complex shapes. Recent advances in monocular depth [40, 5, 21, 59, 13, 54] and geometry prediction [55] allows for reliable geometry estimation from a single image, which can be leveraged for imparting 3D-awareness in into SC methods.

Concurrent work [52] also proposes to geometrically align images in 3D using depth maps to build category prototypes. It is designed around a test-time alignment of the 3D prototype to the test image. This requires knowing the test category beforehand, having build a dedicated prototype for it, having a segmentation mask for the test instance, and solving a computationally expensive alignment problem, limiting its applicability to severely constrained scenarios where the category has been seen during training and throughput it not an issue. In comparison, while our model shares a similar concept of building 3D category prototype, our goal is to design a generalizable SC pipeline by training a correspondence head on top of a backbone. This allows our model to compute correspondence in a single feedforward pass and generalize to new categories all the while not requiring knowledge of the test category or segmentation.

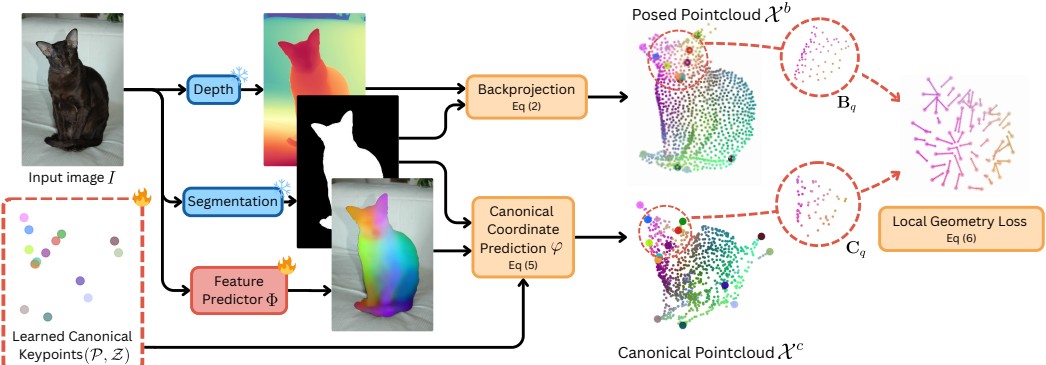

Figure 2: **Overview of our approach.** We extract segmentation masks and depths maps from training images and backproject object points to produce the posed point clouds $\mathcal{X}^b$. We predict dense features with $\Phi$ and match them against our jointly learned sparse category prototype $(\mathcal{P}, \mathcal{Z})$ to produce the canonical point clouds $\mathcal{X}^c$. The local geometric alignment between the two provides supervision for updating $\Phi$.

# 3 Method

## 3.1 Overview

Let $I \in \mathbb{R}^{H \times W \times 3}$ be an RGB image depicting an object, defined over the image domain $\Lambda \in \mathbb{R}^2$, a lattice of size $H \times W$. Our objective is to learn a function $\Phi(I, \boldsymbol{u}) \rightarrow \boldsymbol{w}$, which maps each pixel coordinate $\boldsymbol{u} \in \Lambda$ to a descriptor $\boldsymbol{w} \in \mathbb{R}^M$. The descriptor $\boldsymbol{w}$ should be semantically consistent, be meaningfully aligned across different images of objects from the same category, and be invariant to changes in pose and shape. Once $\Phi$ is learned, SC between two images $I$ and $I'$, depicting the same object category, can be established by finding, for a pixel $\boldsymbol{u}$ in image $I$, the most similar pixel $\boldsymbol{u}'$ in the other image $I'$. This is done by querying nearest-neighbor matching in descriptor space according to distance $d$, typically the cosine distance, i.e., $d(\boldsymbol{a}, \boldsymbol{b}) = 1 - \langle \boldsymbol{a}, \boldsymbol{b} \rangle / (||\boldsymbol{a}||_2 ||\boldsymbol{b}||_2)$:

$$\boldsymbol{u}' = \arg \min_{\boldsymbol{v}} d(\Phi(I, \boldsymbol{u}), \Phi(I', \boldsymbol{v})). \tag{1}$$

In the standard supervised SC task, we are given a training set, $\{(I^{(1)}, \mathcal{K}^{(1)}), \ldots, (I^{(N)}, \mathcal{K}^{(N)})\}$ where each image $I^{(n)}$ is annotated with a sparse set of semantic keypoints $\mathcal{K} = \{\boldsymbol{k}_1, \ldots, \boldsymbol{k}_{|\mathcal{K}|}\}$ where $\boldsymbol{k} \in \Lambda$. A common strategy, adopted in recent works [18, 8, 9, 31, 61, 62], is to learn a descriptor function $\Phi(I, \boldsymbol{u})$ that produces a local descriptor $\boldsymbol{w}$ for each pixel $\boldsymbol{u}$ in $I$ such that the descriptors of corresponding keypoints in paired images are close in feature space. While this sparse keypoint supervision helps the model learn semantically meaningful descriptors for the annotated keypoints, it does not guarantee generalization to unlabeled regions of the object.

A promising direction to address this limitation is to incorporate 3D geometry by assigning each pixel a coordinate in an object centric reference frame. Prior works [46, 32] explores this idea by projecting object surfaces onto a spherical coordinate system, with each coordinate on the sphere corresponding to a different characteristic point of the object. However, this necessitates inferring both object shape and the viewpoint from a collection of 2D images, a highly ill-posed problem, which requires generating pairs through synthetic warps resulting unrealistic shapes [46] or relies on viewpoint supervision which can be challenging to predict automatically. In the next section we show how to combine sparse keypoints annotations with 3D geometry cues to learn dense and semantically consistent descriptors for every pixel in an image. An overview of our approach is shown in Fig. 2.

## 3.2 Canonical Representation Learning

Similar to [46, 32], we aim to learn a 3D *canonical* representation for each object category, along with a function $\varphi(I, \boldsymbol{u}) \rightarrow \boldsymbol{x} \in \mathbb{R}^3$ that maps a pixel $\boldsymbol{u}$ in image $I$ to its 3D coordinates in the canonical object-centric coordinate system. Unlike the spherical representations in prior work [46, 32], we do not impose any topological constraints on the canonical representation (e.g., enforcing a spherical surface). We parameterized the canonical representation by a set of 3D keypoints $\mathcal{P} = \{\boldsymbol{p}_1, \ldots, \boldsymbol{p}_{|\mathcal{K}|}\}$, where each $\boldsymbol{p}_i$ corresponds to a labeled keypoint $\boldsymbol{k}_i$ in $I$. A crucial aspect of this parametrization is

that unlike the labeled image keypoints $\mathcal{K}$, $\mathcal{P}$ is shared across all instances of the category, and is invariant to object instance, viewpoint, and pose, ensuring the canonicity of the representation.

To compute $\mathcal{P}$, we first estimate the 3D coordinates of each keypoint $\boldsymbol{k}$ in image $I$ using a monocular depth model $\Psi(I, \boldsymbol{k}) \to \mathbb{R}^+$ and then backproject it using estimated camera intrinsics $\mathbf{A} \in \mathbb{R}^{3 \times 3}$ as follows:

$$\bar{\boldsymbol{k}} = \Psi(I, \boldsymbol{k})\mathbf{A}^{-1}\left[k_x, k_y, 1\right]^\top, \tag{2}$$

where $\boldsymbol{k} = (k_x, k_y)$ and $\bar{\boldsymbol{k}} \in \mathbb{R}^3$ represents the 3D 'posed' coordinate. We denote the set of backprojected coordinates as $\bar{\mathcal{K}} = \{\bar{\boldsymbol{k}}_1, \ldots, \bar{\boldsymbol{k}}_{|\mathcal{K}|}\}$.

To align $\bar{\mathcal{K}}$ with $\mathcal{P}$, we compute a rigid transformation, comprising rotation $\mathbf{R} \in SO(3)$ and translation $\mathbf{T} \in \mathbb{R}^{3 \times 1}$, and scale $s \in \mathbb{R}^+$ such that $\mathbf{M} = s[\mathbf{R}|\mathbf{T}] \in \mathbb{R}^{3 \times 4}$. We optimize the canonical keypoints $\mathcal{P}$ by minimizing their alignment error across the training set by solving a generalized Procrustes problem:

$$\min_{\boldsymbol{p}_i} \sum_{n=1}^{N} ||\boldsymbol{p}_i - \hat{\boldsymbol{M}}^{(n)}\bar{\boldsymbol{k}}_i^{(n)}||_1 \quad \text{where} \quad \hat{\boldsymbol{M}}^{(n)} = \arg\min_{\mathbf{M}} \sum_{i=1}^{|\mathcal{K}|} ||\boldsymbol{p}_i - \mathbf{M}\bar{\boldsymbol{k}}_i^{(n)}||_2. \tag{3}$$

We use the Kabsch-Umeyama algorithm [19, 50] to compute the optimal transformation $\hat{\boldsymbol{M}}^{(n)}$ between the canonical and posed coordinates keypoints. To prevent the degenerate solution where $s = 0$ leads to a global collapse, we modify the procedure to constrain $s \geq 1$. This ensures objects cannot shrink in size, effectively resizing them to the size of the largest object in the training set. Even though alignments rely on only a sparse subset of visible keypoints per-image, i.e., occluded and out-of-frame points are not considered, we find this sufficient to recover a globally consistent arrangement for $\mathcal{P}$.

Next we associate each canonical keypoint in $\boldsymbol{p}$ with a learnable descriptor $\boldsymbol{z}$, forming a set $\mathcal{Z} = \{\boldsymbol{z}_1, \ldots, \boldsymbol{z}_{|\mathcal{K}|}\}$. We learn these jointly with $\Phi$, using a cross-entropy loss over cosine similarities between extracted and canonical descriptors:

$$\min_{\Phi, \mathcal{Z}} -\frac{1}{N} \sum_{n=1}^{N} \sum_{i=1}^{|\mathcal{K}|} \log \frac{\exp(\text{sim}(\boldsymbol{z}_i, \Phi(I^{(n)}, \boldsymbol{k}_i^{(n)})/\tau)}{\sum_{j=1}^{|\mathcal{K}|} \exp(\text{sim}(\boldsymbol{z}_j, \Phi(I^{(n)}, \boldsymbol{k}_i^{(n)})/\tau)}, \tag{4}$$

with cosine similarity $\text{sim}(\cdot, \cdot)$ and learned temperature parameter $\tau$. This objective encourages $\Phi$ to produce distinctive and semantically consistent features across object instances for each keypoint $\boldsymbol{k}$.

### 3.3 Dense Geometric Alignment

So far, we have only modeled object geometry at the sparse level via $\mathcal{P}$. We now extend this to dense correspondence by defining $\varphi(I, \boldsymbol{u})$, a function that maps every pixel $\boldsymbol{u}$ in $I$ to a coordinate in the canonical space. We compute it as an attention-weighted sum over canonical keypoints:

$$\varphi(I, \boldsymbol{u}) = \sum_{i=1}^{|\mathcal{K}|} \frac{\exp(\text{sim}(\boldsymbol{z}_i, \Phi(I, \boldsymbol{u}))/\tau)}{\sum_{j=1}^{|\mathcal{K}|} \exp(\text{sim}(\boldsymbol{z}_j, \Phi(I, \boldsymbol{u}))/\tau)} \boldsymbol{p}_i. \tag{5}$$

This is equivalent to computing descriptors via softmax attention over $\mathcal{Z}$, using $\Phi(I, \boldsymbol{u})$ as queries, $\boldsymbol{z}$ as keys, and $\boldsymbol{p}$ as values. For labeled keypoints $\boldsymbol{k}_l$, minimizing Eq. (4) ensures $\varphi(I, \boldsymbol{k}_l) = \boldsymbol{p}_l$.

For each training image $I$, we can now estimate the dense canonical coordinates $\mathcal{X}^c$ over its pixels via Eq. (5), and the posed coordinates $\mathcal{X}^b$ via depth backprojection using Eq. (2). In practice, $\mathcal{X}^c$ and $\mathcal{X}^b$ only consist of object points that are selected using an object segmentation mask. We aim to align these two representations so that $\mathcal{X}^c$ properly reflects the object geometry. However, our annotations are only sparse, thus we cannot directly supervise $\varphi(I, \boldsymbol{u})$ for arbitrary coordinates $\boldsymbol{u}$. Instead, we make the assumption that even though the posed and canonical shape are different, they should be *locally* similar. We encourage the geometric alignment between a small neighborhood of points sampled in the posed space, and their corresponding locations in the canonical space.

For a given point $\mathbf{q} \in \mathcal{X}^c$, we sample its $k$ nearest neighbors to obtain $\mathbf{C}_q$ in the canonical space, and the corresponding coordinates in the posed space $\mathbf{B}_q$, and minimize the alignment error between

two sets. We also sample neighbors of a given point $r \in \mathcal{X}^b$ in the posed space, denoted as $\mathbf{B}_r$, and compute the loss in the other direction:

$$\min_{\Phi, \mathcal{Z}} \frac{1}{N} \sum_{n=1}^{N} ||\mathbf{C}_q^{(n)} - \mathbf{M}_{c2b}^{(n)} \mathbf{B}_q^{(n)}||_1 + ||\mathbf{B}_r^{(n)} - \mathbf{M}_{b2c}^{(n)} \mathbf{C}_r^{(n)}||_1, \qquad (6)$$

where $\mathbf{M}_{c2b}$ and $\mathbf{M}_{b2c}$ are the rigid transformations between the canonical and posed coordinates, computed using the Kabsch-Umeyama algorithm at each iteration as in Eq. (3).

In the canonical space, neighbors are selected using a standard k-nearest neighbors strategy. However, this approach can be unreliable in the posed space due to object deformations. For instance, in the case of a person eating, the hand might be close to the face in 3D space, and thus points belonging to the face might mistakenly be selected as neighbors of the hand. Instead, we use a pseudo-geodesic sampling strategy that samples points along the surface of the object. Starting from a seed point, we iteratively grow the neighborhood by selecting the next point with the shortest distance to the current set, effectively approximating surface-based proximity rather than raw spatial closeness. Pseudocode is provided in Alg. 1.

We jointly optimize the descriptor learning loss in Eq. (4) and the geometric consistency loss in Eq. (6) to learn $\Phi$ and $\mathcal{P}$. While these objectives suffice to learn a SC model, in practice we build our implementation on Geo-SC [62] and optimize its parameters jointly over the sum of our objective and the original one. At inference time, rather than simply querying nearest-neighbor in the descriptor space predicted by $\Phi$, we make use of the soft-argmax window matching strategy proposed in [62]. Unlike $\varphi$, which relies on category-specific canonical coordinate set $\mathcal{P}$ and descriptors $\mathcal{Z}$, $\Phi$ can be applied to previously unseen object categories directly.

## 4  SPair-U: A Benchmark for Evaluating Unseen Keypoints

As illustrated in Fig. 1, the performance of the state-of-the-art supervised SC methods [62, 61] degrades significantly when queried on keypoints that are not part of their training sets. We posit that this is caused by models only learning strong representations for these specific points, while largely ignoring the remaining pixels. We would like to assess the performance of SC methods when evaluated on keypoints that were previously unseen (i.e., not in the labeled set) at training time. A possible solution is to use an existing dataset while splitting the annotations into two mutually exclusive sets of keypoints, seen and unseen, between training and evaluation. However, this strategy would reduce the supervision available, and require retraining previous techniques for evaluation.

Instead, we introduce a new evaluation benchmark, **SPair-U**, by labeling additional keypoints from the SPair-71k dataset [34]. We added at least four new points for each of the 18 categories found in SPair-71k. For animals, we focused on additional joints on the limbs, and for vehicles we added semantic parts that were not already labeled, e.g., windshield or fenders. Boats, bottles, potted plants, and tv monitors keypoints are not semantic *per se* in SPair-71k, but are rather spread around on the outline of the object. Thus, we added midway points between those already defined. In total, we add 1,272 new individual test keypoint annotations resulting in 19,990 new keypoint pairs spread across 8,254 image pairs. We illustrate some of these new annotations in Fig. 3, and the full list with more details can be found in Table A3. As shown in Fig. 1, current supervised methods tend to predict locations of keypoints seen during when queried on the new SPair-U points.

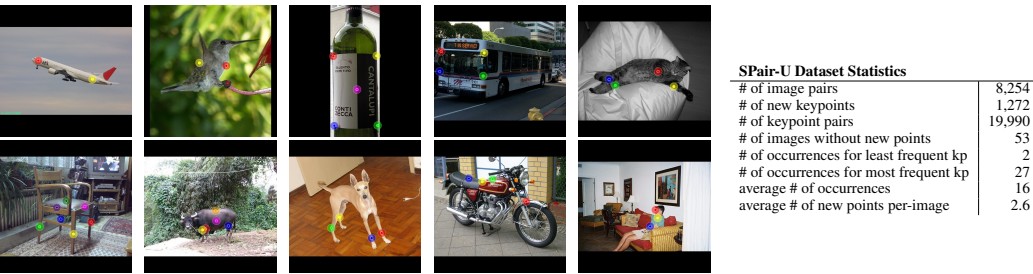

| SPair-U Dataset Statistics | |
| --- | --: |
| # of image pairs | 8,254 |
| # of new keypoints | 1,272 |
| # of keypoint pairs | 19,990 |
| # of images without new points | 53 |
| # of occurrences for least frequent kp | 2 |
| # of occurrences for most frequent kp | 27 |
| average # of occurrences | 16 |
| average # of new points per-image | 2.6 |

Figure 3: **Example keypoint annotations from our new SPair-U evaluation dataset.** It utilizes the same images as the SPair-71k dataset [34], but adds additional keypoints not present in SPair-71k. This enables benchmarking of SC methods on the existing keypoints along with our new ones. On the right we summarize the main statistics of our new dataset.

Table 1: **Results on standard evaluation keypoints for SPair-71k**. Per-image PCK@$0.1_{bbox}$ scores are reported. In this table and the following: All models use the soft matching strategy described in [62] except those followed by *. Models with a dagger[†] benefit from AP-10K pretraining. Models in the $\mathcal{K}$ category use keypoint supervision, while $\not\mathcal{K}$ do not. Best results are **bolded** and second best are underlined.

| | aero | bike | bird | boat | bottle | bus | car | cat | chair | cow | dog | horse | mbike | person | plant | sheep | train | tv | avg |
|---|---|---|---|---|---|---|---|---|---|---|---|---|---|---|---|---|---|---|---|
| $\not\mathcal{K}$ SD [41][61] | 62.8 | 52.7 | 80.6 | 31.2 | 43.4 | 39.1 | 35.6 | 76.0 | 32.0 | 67.6 | 50.9 | 59.7 | 51.0 | 47.3 | 48.6 | 43.8 | 61.8 | 52.9 | 52.0 |
| DINOv2 [39][61] | 73.4 | 60.2 | 88.8 | 43.2 | 41.1 | 46.7 | 45.1 | 75.0 | 33.4 | 69.8 | 66.1 | 69.6 | 60.7 | 66.6 | 30.7 | 61.3 | 54.2 | 23.9 | 55.3 |
| DINOv2+SD [61] | 73.8 | 61.0 | 89.6 | 40.2 | 52.5 | 47.4 | 44.1 | 81.1 | 41.5 | 76.8 | 64.8 | 70.5 | 61.7 | 66.3 | 54.3 | 62.7 | 63.5 | 52.4 | 61.1 |
| SphericalMaps [32] | 76.2 | 60.1 | 90.0 | 46.5 | 53.0 | 74.9 | 68.0 | 83.8 | 45.1 | 81.7 | 67.6 | 75.4 | 69.1 | 58.9 | 50.0 | 67.5 | 73.9 | 58.1 | 66.1 |
| $\mathcal{K}$ SCorrSan* [18] | 57.1 | 40.3 | 78.3 | 38.1 | 51.8 | 57.8 | 47.1 | 67.9 | 25.2 | 71.3 | 63.9 | 49.3 | 45.3 | 49.8 | 48.8 | 40.3 | 77.7 | 69.7 | 55.3 |
| CATS++* [9] | 60.6 | 46.9 | 82.5 | 41.6 | 56.8 | 64.9 | 50.4 | 72.8 | 29.2 | 75.8 | 65.4 | 62.5 | 50.9 | 56.1 | 54.8 | 48.2 | 80.9 | 74.9 | 59.8 |
| DHF* [31] | 74.0 | 61.0 | 87.2 | 40.7 | 47.8 | 70.0 | 74.4 | 80.9 | 38.5 | 76.1 | 60.9 | 66.8 | 66.6 | 70.3 | 58.0 | 54.3 | 87.4 | 60.3 | 64.9 |
| DINO+SD (S) [61] | 84.7 | 67.5 | 93.2 | 64.5 | 59.2 | 85.7 | 82.0 | 89.8 | 57.0 | 89.3 | 76.2 | 80.8 | 75.9 | 80.2 | 64.7 | 71.2 | 93.6 | 70.5 | 76.5 |
| Geo-SC [62] | 86.6 | 70.7 | 95.8 | 69.2 | 64.8 | 94.5 | 90.6 | 91.0 | 67.1 | 91.8 | 86.1 | 86.3 | 79.3 | 87.9 | 80.8 | 82.1 | **96.6** | 83.4 | 83.2 |
| Geo-SC[†] [62] | 92.0 | 76.1 | **97.2** | 70.4 | **70.5** | 91.4 | 89.7 | 92.7 | 73.4 | **95.0** | 90.5 | **87.7** | 81.8 | 91.6 | **82.3** | **83.4** | 96.5 | **85.3** | **85.6** |
| Ours | 86.8 | 72.6 | 95.3 | 70.7 | 64.8 | 94.6 | 90.3 | 89.4 | 70.7 | 94.1 | 84.8 | 83.0 | 80.5 | 87.0 | 79.1 | 77.5 | 95.8 | 82.8 | 82.9 |
| Ours[†] | **92.2** | **76.3** | 96.5 | **72.0** | 68.1 | **95.0** | **90.8** | **93.1** | **75.1** | 94.2 | **91.2** | 86.0 | **82.1** | **91.7** | 80.0 | 81.2 | 95.8 | 84.0 | 85.4 |

Table 2: **Results on unseen keypoints on our SPair-U benchmark.** Per-image PCK@$0.1_{bbox}$ scores on unseen keypoints are reported.

| | aero | bike | bird | boat | bottle | bus | car | cat | chair | cow | dog | horse | mbike | person | plant | sheep | train | tv | avg |
|---|---|---|---|---|---|---|---|---|---|---|---|---|---|---|---|---|---|---|---|
| $\not\mathcal{K}$ SD [41][61] | 73.2 | 71.8 | 48.8 | 37.7 | 43.0 | 55.1 | 47.2 | 25.4 | 35.9 | 60.4 | 46.2 | 41.6 | 59.9 | 53.1 | 57.8 | 36.1 | 50.6 | 19.5 | 47.4 |
| DINOv2 [39][61] | 88.2 | 75.6 | **79.0** | 52.9 | 39.8 | 54.1 | 60.0 | 43.9 | 34.8 | 67.2 | 64.6 | 53.6 | **75.8** | 79.1 | 37.8 | 45.6 | 53.3 | 8.4 | 54.9 |
| DINOv2+SD [61] | 88.0 | **80.4** | 72.3 | 48.2 | **47.9** | 62.3 | 61.5 | 44.8 | 45.0 | 73.0 | 64.7 | 58.2 | 75.5 | **80.0** | 62.7 | 46.1 | **55.9** | 16.9 | 59.4 |
| SphericalMaps [32] | **90.2** | 76.8 | 71.7 | 55.6 | 44.6 | 89.5 | **81.7** | 50.8 | 46.4 | 71.2 | 70.4 | 62.9 | 65.4 | 68.2 | 56.1 | 45.9 | 51.6 | 26.9 | 61.0 |
| $\mathcal{K}$ SCorrSan* [18] | 56.9 | 26.9 | 23.0 | 37.6 | 31.4 | 52.8 | 41.7 | 16.6 | 15.4 | 21.0 | 47.1 | 17.8 | 27.3 | 48.1 | 47.8 | 20.1 | 28.0 | **34.2** | 32.7 |
| CATS++* [9] | 69.9 | 43.8 | 14.0 | 47.1 | 31.9 | 69.5 | 47.0 | 11.7 | 24.4 | 15.1 | 48.2 | 32.0 | 54.3 | 51.6 | 17.5 | 27.9 | 22.8 | 35.9 | 35.9 |
| DHF* [31] | 71.4 | 58.1 | 39.1 | 35.8 | 44.7 | 74.0 | 40.2 | 33.5 | 27.4 | 52.0 | 50.4 | 41.6 | 56.5 | 51.6 | 41.6 | 30.0 | 42.5 | 14.5 | 43.3 |
| DINO+SD (S) [61] | 81.5 | 73.6 | 57.1 | 63.4 | 35.8 | 85.7 | 67.7 | **64.3** | 39.3 | 67.9 | **86.8** | **79.5** | 60.9 | 70.1 | 55.8 | **57.8** | 42.7 | 12.6 | 60.0 |
| Geo-SC [62] | 80.9 | 71.4 | 51.8 | 65.3 | 36.9 | 91.0 | 70.8 | 55.7 | 36.9 | 55.7 | 79.2 | 53.7 | 66.5 | 62.3 | 61.1 | 39.0 | 17.4 | | 56.9 |
| Geo-SC[†] [62] | 74.6 | 70.6 | 55.5 | 65.1 | 36.4 | 85.1 | 72.3 | 50.1 | 40.1 | 60.6 | 85.3 | 65.7 | 52.9 | 61.9 | 66.6 | 41.8 | 36.6 | 13.8 | 57.1 |
| Ours | 80.3 | 74.5 | 70.6 | 67.1 | 40.2 | **92.9** | 72.7 | 53.8 | 45.8 | 68.5 | 75.3 | 62.0 | 67.8 | 65.4 | 68.1 | 45.4 | 47.9 | 30.5 | 62.4 |
| Ours[†] | 81.1 | 73.2 | 72.0 | **67.5** | 35.2 | 92.1 | 75.5 | 61.2 | **51.4** | **74.3** | **86.8** | 78.8 | 70.9 | 68.9 | **72.6** | 54.7 | 44.8 | 32.2 | **66.1** |

# 5 Experimental Results

## 5.1 Implementation Details

Our 3D prototype approach is complementary to existing semantic correspondence architectures, thus we can add it as an additional objective on top of established models. We base our experiments on Geo-SC [62], strictly following their provided hyperparameters, e.g., learning rate, batch size, optimizer, scheduler, and epoch count, simply adding our additional loss terms and jointly optimizing $\mathcal{P}$ and $\mathcal{Z}$ alongside Geo-SC's feature extractor $\Phi$. We also preserve the contrastive $\mathcal{L}_{sparse}$ and dense $\mathcal{L}_{dense}$ objectives with gaussian noise, as well as feature maps dropout and pose-variant augmentation. We refer to the original publication and official implementation for in-depth description of these features. While $\mathcal{P}$ and $\mathcal{Z}$ are category-specific, a single $\Phi$ is trained on the full dataset, allowing generalization to new categories. Furthermore, gradients coming from Eq. (6) are not backpropagated to $\mathcal{P}$, meaning it is only optimized using Eq. (3).

Our complete loss term is $\mathcal{L}_{sparse} + \mathcal{L}_{dense} + \mathcal{L}_{\mathcal{P}} + 0.3 \times \mathcal{L}_{\mathcal{Z}} + \mathcal{L}_{geom}$, where $\mathcal{L}_{\mathcal{P}}$, $\mathcal{L}_{\mathcal{Z}}$, and $\mathcal{L}_{geom}$ correspond to Eq. (3), Eq. (4) and Eq. (6) respectively. The justification for setting the weight $\lambda_{\mathcal{Z}} = 0.3$ is provided in Appendix B.

In practice, $\Phi$ consists of additional bottleneck layers trained on top of frozen DINOv2 and SD backbones. We extract depth maps and camera intrinsics using MoGe [54], and use Segment Anything [23] to obtain segmentation masks. Importantly, these are only used during training. During evaluation, matches are computed only from the predictions of $\Phi$. Additional implementation details can be found in Appendix A.

## 5.2 Quantitative Results

**Seen – SPair-71k.** We first compare our models to other SC approaches on the SPair-71k benchmark [34], which contains images from 18 categories. We use the standard PCK@$0.1_{bbox}$ which considers a match to be correct if its prediction lies within distance $0.1 \times \max(h, w)$ of the ground truth location, where $(h, w)$ is the height and width of the target object bounding box. Typically, supervised models report *per-image* PCK, i.e., the average score of each image per-category, while unsupervised ones use *per point* PCK, i.e., the average number of correct matches per-category. In order to properly compare results between the two families, we recompute *per-image* PCK for all models, which results in a small drop for the unsupervised models.

Table 3: **Cross-benchmark evaluation on held-out datasets.** Scores are reported using PCK with different thresholds. Here, only keypoint supervision from Spair-71k is used for supervised models.

| | Spair-71k | | | Spair-U | | | AP-10K IS | | | AP-10K CS | | | AP-10K CF | | | PF-PASCAL | | |
|---|---|---|---|---|---|---|---|---|---|---|---|---|---|---|---|---|---|---|
| PCK threshold | 0.01 | 0.05 | 0.10 | 0.01 | 0.05 | 0.10 | 0.01 | 0.05 | 0.10 | 0.01 | 0.05 | 0.10 | 0.01 | 0.05 | 0.10 | 0.05 | 0.10 | 0.15 |
| $\not{K}$ SD [41][61] | 6.3 | 37.3 | 52.0 | 3.3 | 28.0 | 47.4 | 8.4 | 36.9 | 52.5 | 6.6 | 32.6 | 47.9 | 4.3 | 24.6 | 37.6 | 66.1 | 80.0 | 85.3 |
| DINOv2 [39][61] | 6.8 | 38.0 | 55.3 | 3.7 | 32.4 | 54.9 | 10.5 | 44.8 | 63.6 | 8.8 | 41.7 | 61.6 | 7.7 | 34.7 | 52.0 | 62.4 | 78.2 | 83.5 |
| DINOv2+SD [61] | 8.2 | 44.2 | 61.1 | **4.7** | 37.0 | 59.4 | 11.7 | 47.4 | 65.8 | 10.0 | 44.0 | 63.5 | 7.7 | 35.4 | 52.4 | 72.5 | 85.6 | 90.3 |
| SphericalMaps [32] | 8.2 | 47.7 | 66.1 | 4.5 | **38.2** | 61.0 | 12.5 | 48.5 | 66.7 | 10.6 | 44.9 | 63.6 | 8.0 | 35.7 | 52.1 | 74.6 | **88.9** | **93.2** |
| $K$ DINO+SD (S) [61] | 13.0 | 61.6 | 76.5 | 3.6 | 35.9 | 59.3 | 15.1 | 54.3 | **71.7** | 13.6 | 51.1 | 68.7 | 11.0 | 44.0 | 60.4 | 74.5 | 87.4 | 91.1 |
| Geo-SC [62] | 20.0 | **72.2** | **83.2** | 4.6 | 35.5 | 56.9 | **16.6** | **55.8** | 70.5 | **15.2** | 52.4 | 67.7 | **11.9** | 45.9 | 59.6 | 75.3 | 87.0 | 90.7 |
| Ours | **20.5** | 72.1 | 82.9 | 4.2 | 37.8 | **62.4** | 16.5 | **55.8** | 71.3 | 15.1 | 53.0 | 69.0 | 11.2 | 46.1 | 61.1 | **75.8** | 87.5 | 91.2 |

Results on seen keypoints in Table 1 show that our model ranks competitively against other approaches, with a marginal 0.2% performance drop on average against its backbone Geo-SC [62]. Per-category results show small improvements in nine of the categories, the highest one being 3.6% on bus, and small drop on the other nine, the largest being 2.4% on bottle. Overall, the differences are minor, illustrating that adding our extra objective does not interfere with the original model.

**Unseen – SPair-U.** To evaluate a model's ability to generalize to unseen semantic points, we assess its performance on our new SPair-U keypoints using per-image PCK@$0.1_{bbox}$ for a like-for-like comparison. We exclude the Test-time Adaptive Pose Alignment from [62] since it requires prior knowledge of keypoint semantics to relabel flipped keypoints, which contradicts the assumption that evaluation keypoints are unknown.

As shown in Table 2, results on SPair-U reveal a stark contrast between supervised and unsupervised models. While unsupervised models see only a modest performance drop, likely due to increased task difficulty, supervised models experience a significant decline. Many of the pre-existing approaches are outperformed by the unsupervised DINO+SD baseline [61] and they are consistently beaten by the weakly-supervised Spherical Maps [32]. Notably, in eight categories, the best-performing model Has not seen keypoints during training, suggesting that supervised approaches behave more like keypoint regressors and fail to generalize to novel correspondences.

Our method also shows some performance degradation on SPair-U, but the drop is smaller than that of its backbone Geo-SC. It achieves the highest overall performance, improving upon the best prior supervised model by 6.1%, indicating stronger generalization to unseen keypoints. Nevertheless, the substantial gap between results on SPair-71k and SPair-U underscores a broader limitation: despite recent progress, most models struggle to move beyond sparse keypoint supervision toward robust, general semantic correspondence.

**Cross-benchmark evaluation.** We further evaluate our model on four benchmarks: SPair-71k, SPair-U, AP-10k [62], and PF-PASCAL [15]. While most supervised SC methods train separate models for each benchmark, this setup encourages overfitting to the benchmark and is impractical for real-world use. Instead, we advocate for evaluating generalization by training a single model on one dataset and testing it across multiple benchmarks. We choose SPair-71k for training due to its balanced mix of object and animal categories, making it suitable for generalization. To ensure fairness, we exclude models pretrained on AP-10K and standardize evaluation using the windowed soft-argmax protocol from [62].

As shown in Table 3, while Geo-SC [62] achieves the best performance on the standard SPair-71k test set, it underperforms on all other benchmarks, highlighting its limited generalization. Notably, even a simple supervised DINO+SD [61] baseline outperforms Geo-SC at the standard 0.10 threshold when using the same soft window matching strategy. This stands in sharp contrast to the findings in [62] where Geo-SC consistently outperforms its baseline by 10% on the three AP-10K benchmarks, when both models are trained on AP-10K, indicating potential overfitting to that dataset.

Consistent with earlier observations, a clear pattern emerges: models trained without keypoint supervision maintain stable rankings and performance gaps across datasets, whereas supervised models cluster more tightly in performance when evaluated out of their training distribution, revealing weaker cross-set generalization.

## 5.3 Qualitative Results

In Fig. 4, we visualize PCA projections of object features produced by our model, Geo-SC, and the unsupervised DINO+SD. Our model produces descriptors that vary smoothly over the object surface while uniquely identifying each point. In comparison, the predictions of Geo-SC are noisier, with

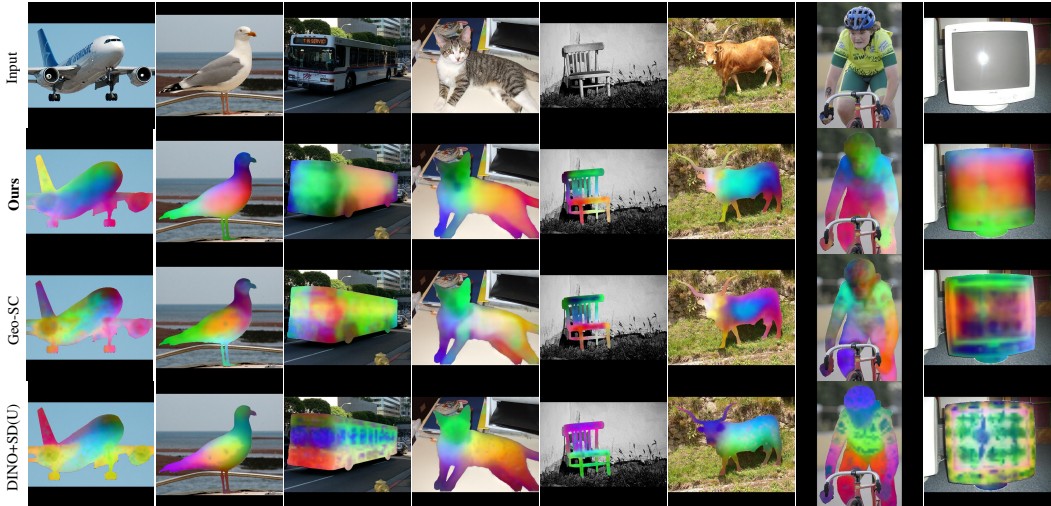

Figure 4: **PCA visualization of the feature maps from different models.** Note that PCA is computed on object features only. The inclusion of geometric constraints during training results in fewer high frequency artifacts in the predicted feature maps for our approach.

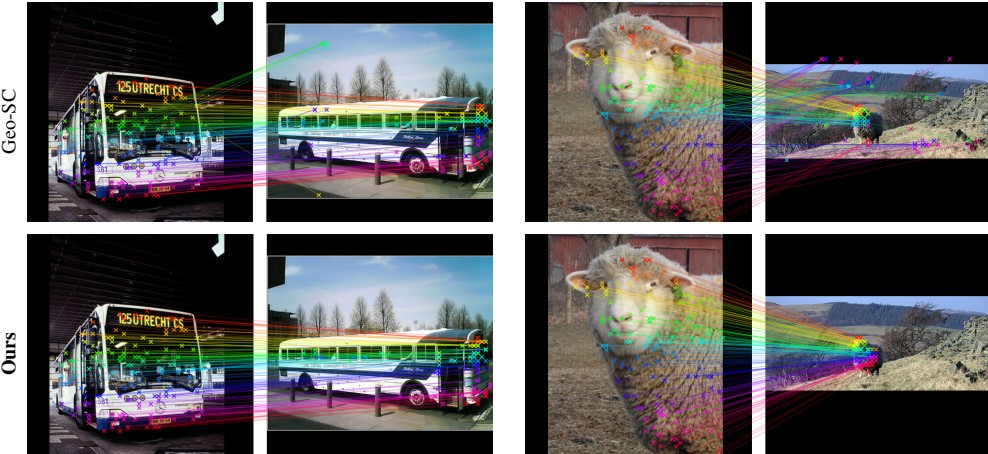

Figure 5: **Visualization of keypoint matches for randomly selected object points.** On each source image (left) we randomly sample points on the object of interest and compute their match on the target (right). Colored lines are used as a way to distinguish the points.

sudden discontinuities (e.g., bus) and uniform descriptors on regions that have no keypoints (e.g., the body of the cat and the cow). Meanwhile, the unsupervised features fail to separate repeated parts (e.g., plane engines) and produces noisy features in textureless areas (e.g., tv).

In Fig. 5 we further qualitatively evaluate our model's ability to generalize to unseen points. We randomly sample points on the source object and compute their matches on the target. Compared with Geo-SC, our approach exhibits better robustness against matching outside the target object, as well as better spatial awareness of points (e.g., Geo-SC matching points from the bottom of the source bus to the roof of the target), leading to higher matching quality. Further examples, along with comparisons to the unsupervised DINOv2+SD backbone, are provided in Fig. A4.

Following [52], we visualize our learned canonical shapes in Fig. 6 by collecting predicted canonical coordinates of multiple objects in order to overlap their partial point clouds over the training data. We observe that the spatial organization of $\mathcal{P}$, i.e., the large bold points, captures the general shape of the category, and the predicted coordinates densely span the object surface. Interestingly, our parametrization of the canonical shape Eq. (5) forces predicted coordinates to lie within the convex hull of $\mathcal{P}$, which explains the incomplete wheel on the motorbike. We also observe that very few points are mapped towards the end of the train, which we attribute to the varying length of trains across instances and the bias toward frontal viewpoints. Note that contrary to [52], these are simply visualizations and are not used for inference, meaning this limitation is unlikely to significantly affect performance.

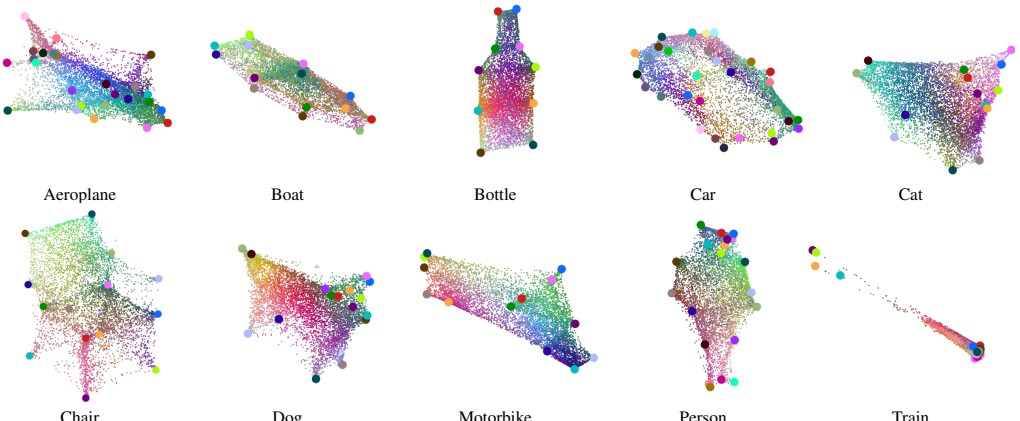

Figure 6: **Visualization of our learned canonical shapes.** Large points correspond to $\mathcal{P}$, each being attributed a distinctive color for visualization. Small points are predicted canonical coordinates of objects, colored with PCA of the features predicted by $\varphi$.

## 6 Limitations

While augmenting the test set of SPair-71k with new keypoints enables us to evaluate existing techniques with their provided models and code, our proposed benchmark SPair-U inherits some drawbacks from SPair-71k. In particular, the test set is small, consisting of only 481 images but with over 8,000 test pairs, and the categories are restricted to only common objects and animals. Furthermore, some categories were already labeled with a high number of keypoints where it is potentially easier to detect the newly added ones by relating them to the existing ones. While our findings related to generalization issues in supervised SC techniques remain valid, a larger, higher-quality held-out set of images and keypoints would be beneficial for more extensive evaluation.

Compared to prior supervised methods [18, 8, 31, 62], our approach incorporates additional supervision in the form of depth maps and segmentation masks, similar to [52], although in our case, they are only used during training. Furthermore, unlike [32], which relies on camera viewpoint annotations that off-the-shelf models cannot reliably provide, we obtain all additional signals using existing pretrained models.

In Eq. (3), we assume there exist a *global* rigid transformation between the posed keypoints and their canonical counterpart, which in practice is not the case especially for deformable objects. However, the sole purpose of this step is to optimize $\mathcal{P}$ into a coarse spatial organization of 3D keypoints (e.g., making sure that the left hand keypoint generally sits opposite of the right one) in order to allow the computation of *local* geometric alignment in Eq. (6). We show in Fig. 6 and Section 5.2 that despite this coarse assumption, our method is able to recover a reasonable 3D structure and performs well on deformable objects.

Finally, our assumption that geometry is a good proxy for semantics breaks down for complex object categories with diverse spatial part configurations. For example, cabinets may have different numbers of doors that open in various directions, leading to inconsistent placement of features like handles. Finally, we do not foresee any negative social impacts of our work.

## 7 Conclusion

We addressed the challenge of estimating semantic correspondences across different image instances of the same object category. Although recent supervised methods perform well on keypoints seen during training, we show that they often struggle to generalize to unseen keypoints. To overcome this, we introduced a new approach that incorporates geometric constraints during training by learning a continuous canonical manifold specific to each category. Our method outperforms both supervised and unsupervised baselines, as demonstrated on SPair-U – a new dataset we introduce with additional keypoint annotations for the widely used SPair-71k benchmark.

**Acknowledgements.** HB was supported by the EPSRC Visual AI grant EP/T028572/1.

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

# Appendix

## A    Additional Implementation Details

**General implementation.** We base our model on Geo-SC [62], reusing all default hyperparameters that come with the official implementation[1], e.g., training for 2 epochs using AdamW [27] optimizer with $1.25 \times 10^{-3}$ initial learning rate and $1.0 \times 10^{-3}$ weight decay, coupled with one-cycle learning rate scheduler [43], with a batch size of $1$. Every $5,000$ iterations, models are evaluated on the validation split, and the best performing model is retained. For evaluation, unless stated otherwise, a soft-argmax window of size 15 is used.

Experiments were performed on a single NVIDIA RTX 6000 Ada Generation, using pre-extracted DINOv2 and SD feature maps, depth maps, and segmentation masks. Training a model on SPair-71k consumes roughly 4.3GB of VRAM over 8 hours, representing an increased memory cost over Geo-SC's 2.9GB, mainly due to the many $\mathcal{X}_b$ and $\mathcal{X}_c$ we sample, and a doubling of runtime from roughly 4 hours. At inference time however, there is no impact as we estimate matches using features predicted with $\Phi$ in the exact same way Geo-SC does.

**Point cloud sampling.** When backprojecting image points using Eq. (2), we obtain a point cloud whose size depends on the number of visible object pixels. This can cause an imbalance of the samples in the final loss terms, with larger objects contributing more. Furthermore, these point clouds have a very specific grid-like structure inherited from the bitmap format of images, which is dense on surfaces parallel to the image plane and gets sparser as the angle increases. Therefore, we first subsample each training point cloud to a size of $k = 1024$ points using farthest point sampling to obtain a fixed number of well-distributed samples.

When computing the local geometry loss $\mathcal{L}_{\text{geom}}$, we use neighborhoods of size $k' = 64$. An ablation study of this parameter is provided in Table A1c. We also provide in Alg. 1 the pseudocode for the sampling strategy used to obtain $\mathbf{B}_r$ in the posed space.

**Input:** Point cloud $\mathcal{X}$, seed point $p$, number of neighbors $k$
**Output:** neighbor set $B$
```
// Start with the seed
```
$B \leftarrow \{p\}$;
**while** $|B| < k$ **do**
    ```// Filter out already selected points```
    $\mathcal{X}' \leftarrow \mathcal{X} \setminus B$;
    **for** $x$ *in* $\mathcal{X}$' **do**
        ```// Compute the distance to the closest point in B```
        $D_x \leftarrow \min_{y \in B} \|x - y\|_2$;
    **end**
    ```// Add the point with minimal distance to B```
    $B \leftarrow B \cup \{\arg\min_{x \in \mathcal{X}'} D_x\}$;
**end**
**return** $B$;

**Algorithm 1:** Pseudo-geodesic sampling

## B    Ablations

**General ablations.** We perform ablations of our designs in Table A1, and report results on the the SPair-71k [34] validation set which helps us chose the best performing model. It is not possible to ablate individual loss terms as they each have a distinct purpose without which the prototype cannot properly be learned: $\mathcal{L}_{\mathcal{P}}$ optimizes $\mathcal{P}$, $\mathcal{L}_{\mathcal{Z}}$ optimizes $\mathcal{Z}$, and $\mathcal{L}_{\text{geom}}$ provides a dense supervision signal, i.e., a loss for $\Phi(I, \boldsymbol{u})$ when $\boldsymbol{u}$ is an arbitrary object pixel, i.e., not a keypoint. We can however examine the different effect of our and Geo-SC's specific loss terms on both SPair-71k and SPair-U, by comparing a simple supervised approach using a DINOv2 and SD backbone trained only using $\mathcal{L}_{\text{sparse}}$, adding only the Geo-SC specific losses, adding only our canonical prototype losses, and

---

[1]https://github.com/Junyi42/geoaware-sc

Table A1: **Results of different ablations.**

(a) Average PCK@0.1 on SPair-71k test set and SPair-U for different models.

| DINOv2+SD | Geo-SC losses | Canonical prototype | Spair-71k | Spair-U |
|:---:|:---:|:---:|:---:|:---:|
| ✓ | | | 76.5 | 60.0 |
| ✓ | ✓ | | 85.6 | 57.1 |
| ✓ | | ✓ | 75.9 | 66.0 |
| ✓ | ✓ | ✓ | 85.4 | 66.1 |

(b) Average PCK@0.1 on SPair-71k validation set for general ablations.

| Ablation | PCK |
|:---|:---:|
| $\lambda_{\mathcal{Z}} = 1$ | 85.9 |
| $\lambda_{\mathcal{Z}} = 0.1$ | 86.1 |
| K-nn sampling | 85.9 |
| Geodesic sampling | 86.2 |
| Full model | 86.5 |

(c) Average PCK@0.1 on SPair-71k validation set for different neighborhood size.

| Neighborhood size | PCK |
|:---:|:---:|
| 4 | 85.5 |
| 8 | 86.1 |
| 16 | 85.9 |
| 32 | 86.5 |
| 64 | 86.5 |
| 128 | 86.6 |
| 256 | 86.1 |

(d) Average PCK@0.1 on SPair-71k validation set for different rigidity constraints.

| Selected points | PCK |
|:---:|:---:|
| 3 | 86.6 |
| 4 | 86.4 |
| 5 | 86.1 |
| 6 | 86.4 |
| 7 | 86.5 |
| 8 | 86.0 |
| all | 86.5 |

our proposed model that combines them. Results shown in Table A1a show a clear pattern, i.e., adding our canonical prototype loss results in a very small drop in performance on seen keypoints (i.e., SPair-71k), which we attribute to the models having less capacity to fully overfit the training keypoint supervision, but endows them with the ability to generalize much better to unseen points (i.e., SPair-U). Conversely, the Geo-SC losses allow models to perform really well on seen keypoints but its effect on generalization ranges from harmful to null (with vs. without Geo-SC losses). These results also demonstrate that our contributions do not require Geo-SC to work, as they also boost performance of the supervised baseline on unseen keypoints (with vs. without Canonical prototype).

We show that setting $\lambda_{\mathcal{Z}}$ to 1 or 0.1 both negatively affect performance. We believe this is due to the interaction between $\mathcal{L}_{\mathcal{Z}}$ and $\mathcal{L}_{\text{geom}}$, as a high $\lambda_{\mathcal{Z}}$ would push $\Phi$ to collapse towards defaulting to predicting keypoint features $\mathcal{Z}$ for most points, while a weight too low prevents correct prediction on the keypoints. We also test different neighbor sampling strategies for $\mathcal{X}_b$ and $\mathcal{X}_c$, and show that sampling both spaces with either K-nearest neighbor or geodesic sampling is ineffective.

**Neighborhood size in $\mathbf{C}_q$ and $\mathbf{B}_r$.** We experiment with different neighborhood sizes when computing $\mathcal{L}_{\text{geom}}$ and valite the different models on the SPair-71k validation set in Table A1c. Results show little to no effect of the neighborhood size, which is consistent with our previous finding that our losses do not improve performance on semantics points that are present in the training set.

**Number of points in Eq. (3)** To compute $\mathcal{L}_{\mathcal{P}}$, we compute a global rigid transformation between the posed and canonical keypoints, which our qualitative analysis in Fig. 6 and Fig. A1 shows to be reasonable despite being a coarse simplification of the problem. In order to evaluate its impact quantitatively, we evaluate altered version of this procedure where we learn $\mathcal{P}$ not by globally aligning all keypoints but only a randomly sampled local subset of them. Results in Table A1d show that only considering a local neighborhood does not impact the validation performance of our model.

## C   Additional Results

### C.1   Additional Metrics

Multiple recent works pointed out issues with evaluating using PCK, and proposed additional evaluation metrics to address its limitations.

**PCK$^\dagger$ [2]** PCK matches are counted correct even if the prediction lies closer to a keypoint that is not the target, which can lead to high scores when many points are grouped together, even though the system does not distinguish between them. The authors introduce PCK$^\dagger$ which only considers a match correct if it lies within the threshold *and* its closest annotated point is the target.

Table A2: **Evaluation under robust metrics.** All metrics use *per-image* averaging, and all models use window soft-argmax. All models are trained on SPair-71k, and models with a double dagger‡ benefit from AP-10K pretraining. Models in the $\mathcal{K}$ category use keypoint supervision, while $\mathcal{K}$̸ do not. Best results are **bolded** and second best are underlined.

| | Spair-71k KAP | | | Spair-U KAP | | | Spair-71k PCK† | | | Spair-U PCK† | | | Spair-71k GA | | | AP-10K IS GA | | |
| Threshold | 0.01 | 0.05 | 0.10 | 0.01 | 0.05 | 0.10 | 0.01 | 0.05 | 0.10 | 0.01 | 0.05 | 0.10 | 0.01 | 0.05 | 0.10 | 0.01 | 0.05 | 0.10 |
|---|---|---|---|---|---|---|---|---|---|---|---|---|---|---|---|---|---|---|
| $\mathcal{K}$̸ SD [41][61] | 38.2 | 47.5 | 53.0 | 43.4 | 51.6 | 58.5 | 6.3 | 34.4 | 44.4 | 3.3 | 27.7 | 45.4 | 4.2 | 28.3 | 43.3 | 1.3 | 15.3 | 31.0 |
| DINOv2 [39][61] | 37.8 | 47.1 | 52.8 | 43.3 | 52.4 | 60.6 | 6.7 | 34.4 | 46.0 | 3.7 | 32.1 | 52.1 | 3.6 | 26.3 | 43.4 | 2.3 | 25.6 | 47.0 |
| DINOv2+SD [61] | 38.4 | 49.6 | 55.9 | 43.7 | 54.2 | 62.8 | 8.1 | 41.0 | 52.8 | **4.7** | 36.6 | 56.6 | 4.8 | 32.7 | 50.8 | 2.4 | 26.1 | 48.1 |
| SphericalMaps [32] | 38.9 | 51.2 | 58.2 | **44.3** | 55.4 | 64.2 | 8.8 | 44.4 | 57.3 | 4.5 | 37.8 | 58.5 | 5.6 | 37.7 | 58.1 | 2.6 | 28.4 | 51.8 |
| $\mathcal{K}$ DINO+SD (S) [61] | 39.1 | 55.4 | 64.1 | 43.8 | 54.7 | 63.9 | 13.0 | 59.7 | 72.0 | 3.6 | 35.5 | 57.2 | 10.2 | 53.6 | 69.7 | 2.8 | 32.1 | 56.6 |
| Geo-SC [62] | 39.8 | 59.3 | 67.8 | 43.8 | 54.2 | 62.8 | 20.0 | 69.8 | 78.3 | 4.6 | 35.1 | 54.9 | 17.2 | 65.6 | 78.0 | **3.7** | **33.6** | 55.3 |
| Geo-SC‡ [62] | **40.1** | **61.0** | **69.2** | 43.8 | 54.1 | 62.8 | **22.0** | **73.0** | **80.9** | 4.3 | 35.8 | 55.3 | **20.0** | **70.9** | 82.3 | - | - | - |
| Ours | 39.8 | 59.8 | 68.1 | 44.0 | 55.0 | 64.5 | 20.4 | 69.8 | 78.1 | 4.2 | 37.4 | 60.3 | 17.4 | 65.8 | 77.7 | 3.5 | 33.5 | **56.1** |
| Ours‡ | 40.0 | 60.3 | 69.0 | 44.2 | **56.0** | **66.0** | 20.8 | 72.1 | 80.7 | 4.5 | **41.3** | **64.2** | 18.8 | 70.7 | **82.4** | - | - | - |

**KAP [32]** PCK only considers matches when both ground-truth points are visible and does not penalize systems that predict strong similarities for points that do not correspond, for instance between the two opposite sides of a car. KAP reformulates the correspondence evaluation as a binary classification problem between the pixels that are close to the target and those those that are not. Crucially, it penalizes high predictions when a source keypoint is invisible in the target.

**Geo-aware subset (GA) [62]** Finally, [2],[62] and [32] noted that SC pipelines - especially unsupervised ones - often make mistakes because of repeated parts and object symmetries. [62] proposed evaluation on the *Geo-aware* subset of points only, e.g., the points for which there is a symmetric corresponding point.

Results in Table A2 confirm the patterns observed in Section 5.2. For all metrics, supervised models performances drop back down to unsupervised-level or worse when evaluated outside their training labels. Interestingly, KAP scores do not widely vary between supervised and unsupervised models, indicating that supervised models are still likely to predict strong similarity between points when none exists.

## C.2 Additional Visualizations

We visualize more canonical surfaces in Fig. A1. While the shapes are sensible, we observe some limitations in adequately modeling categories with extreme deformations like birds: points belonging to the wings are predicted close to the body when they are folded, and away when they are spread. However, this is consistent with SPair-71k labeling, where the tips are only labeled when the wings are spread.

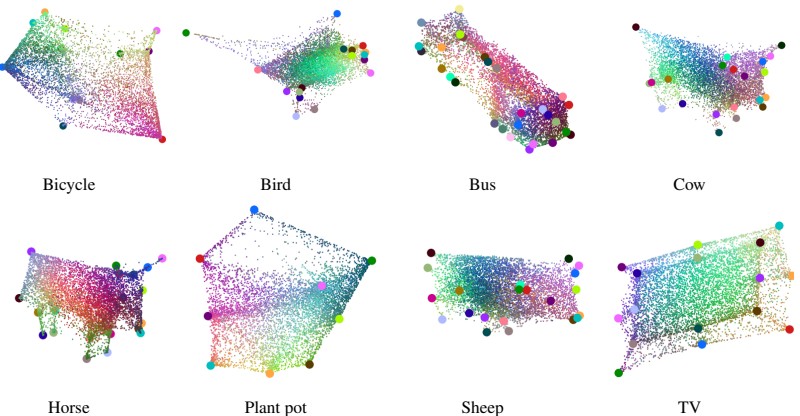

Bicycle          Bird          Bus          Cow

Horse          Plant pot          Sheep          TV

Figure A1: **Visualization of extra canonical shapes.** Large points correspond to $\mathcal{P}$, each being attributed a distinctive color for visualization. Small points are predicted canonical coordinates of objects, colored with PCA of the features predicted by $\varphi$.

We also show some predictions for the unsupervised DINOv2+SD, Geo-SC, and our model on SPair-U in Fig. A3. We observe some interesting failure cases: on the aeroplane, the unsupervised model correctly matches the door, while both supervised models incorrectly predict a training keypoint. In two occasions, Geo-SC predicts points outside of the object when queried on points that are far from training annotations (cow and person). Finally, two very challenging cases are shown with the chair and the tv, illustrating that generic semantic correspondence is still a particularly challenging task.

# D    Additional SPair-U Details

We annotated images using the VGG Image Annotator [12]. We further post-processed the annotations into JSON files replicating the structure of SPair-71k annotations, i.e., per-image annotations and a list of testing pairs. This allows SPair-U to function as a drop-in replacement for SPair-71k evaluation in any semantic correspondence evaluation script. Note that it is designed to be a benchmark of unseen semantic points intended for evaluating the generalization ability of SC models, therefore does not come with a training or validation split. We present the full list of keypoint semantics of SPair-U in Table A3, per-category statistics in Table A4, and some keypoint visualization in Fig. A2.

Table A3: **List of SPair-U keypoint semantics.**

| Category | Keypoint semantics |
|---|---|
| Aeroplane | front-left, front-right, rear-left, rear-right doors |
| Bicycle | top and bottom of head tube; front brake; rear brake |
| Bird | center of back, chest; left wing wrist; right wing wrist |
| Boat | midpoint of the bow; front-left, front-right, rear-left, rear-right side midpoints |
| Bottle | center and corner points of label |
| Bus | top-left, top-right, bottom-left, bottom-right corners of windshield |
| Car | front-left, front-right, rear-left, rear-right top of the wheel arches |
| Cat | front-left, front-right, rear-left, rear-right hocks |
| Chair | leg midpoints; seat edge midpoints; seat center |
| Cow | left and right shoulder joints; left and right hip joints; left and right centers of the body; middle of back |
| Dog | front-left, front-right, rear-left, rear-right hocks |
| Horse | left and right shoulder joints; left and right hip joints |
| Motorbike | front fender midpoint; seat front edge, seat rear edge; engine compartment center |
| Person | forehead center; navel; neck base; left hip joint, right hip joint |
| Plant Pot | center of pot; midpoints of edges; midpoints of rim |
| Sheep | left and right shoulder joints; left and right hip joints |
| Train | locomotive rear top-left, top-right, bottom-left, bottom-right corners |
| Tv | center point; top-left, top-right, bottom-left, bottom-right quadrant centers |

Table A4: **Per-category statistics for our SPair-U benchmark.**

| | ✈ | 🚲 | 🐦 | ⛵ | 🍾 | 🚌 | 🚗 | 🐈 | 🪑 | 🐄 | 🐕 | 🐎 | 🏍 | 🚶 | 🌱 | 🐑 | 🚆 | 🖥 | Avg |
|---|---|---|---|---|---|---|---|---|---|---|---|---|---|---|---|---|---|---|---|
| Image count | 27 | 26 | 27 | 27 | 30 | 27 | 25 | 25 | 26 | 26 | 25 | 25 | 27 | 26 | 30 | 27 | 28 | 27 | 27 |
| Number of pairs | 254 | 576 | 480 | 666 | 338 | 304 | 300 | 510 | 552 | 466 | 488 | 420 | 536 | 488 | 744 | 218 | 314 | 600 | 458 |
| Count of new semantic labels | 4 | 4 | 4 | 5 | 5 | 4 | 4 | 4 | 9 | 7 | 4 | 4 | 4 | 5 | 5 | 4 | 4 | 5 | 4.7 |
| Total labeled points | 39 | 74 | 37 | 74 | 71 | 66 | 44 | 64 | 138 | 81 | 72 | 60 | 67 | 62 | 118 | 39 | 50 | 116 | 70.7 |
| Average number of visible points | 1.4 | 2.9 | 1.4 | 2.7 | 2.4 | 2.4 | 1.8 | 2.6 | 5.3 | 3.1 | 2.9 | 2.4 | 2.5 | 2.4 | 3.9 | 1.4 | 1.8 | 4.3 | 2.6 |
| Number of zero-kp images | 3 | 1 | 2 | 0 | 11 | 9 | 0 | 1 | 1 | 2 | 2 | 3 | 2 | 0 | 2 | 10 | 2 | 2 | 2.9 |
| Min keypoint occurrence | 8 | 14 | 2 | 11 | 13 | 17 | 10 | 14 | 11 | 8 | 17 | 15 | 16 | 9 | 22 | 9 | 12 | 23 | 12.8 |
| Avg keypoint occurrence | 10.8 | 19.5 | 10.3 | 15.8 | 15.2 | 17.5 | 12.0 | 17.0 | 16.3 | 12.6 | 19.0 | 16.0 | 17.8 | 13.4 | 24.6 | 10.8 | 13.5 | 24.2 | 15.9 |
| Max keypoint occurrence | 13 | 25 | 20 | 23 | 19 | 18 | 14 | 21 | 21 | 19 | 20 | 17 | 20 | 21 | 27 | 13 | 15 | 25 | 19.5 |
| Avg kp per pair | 1.4 | 2.4 | 1.2 | 1.7 | 2.9 | 3.4 | 1.5 | 1.9 | 3.8 | 2.0 | 2.5 | 2.0 | 2.0 | 1.7 | 3.6 | 1.6 | 1.9 | 4.3 | 2.3 |

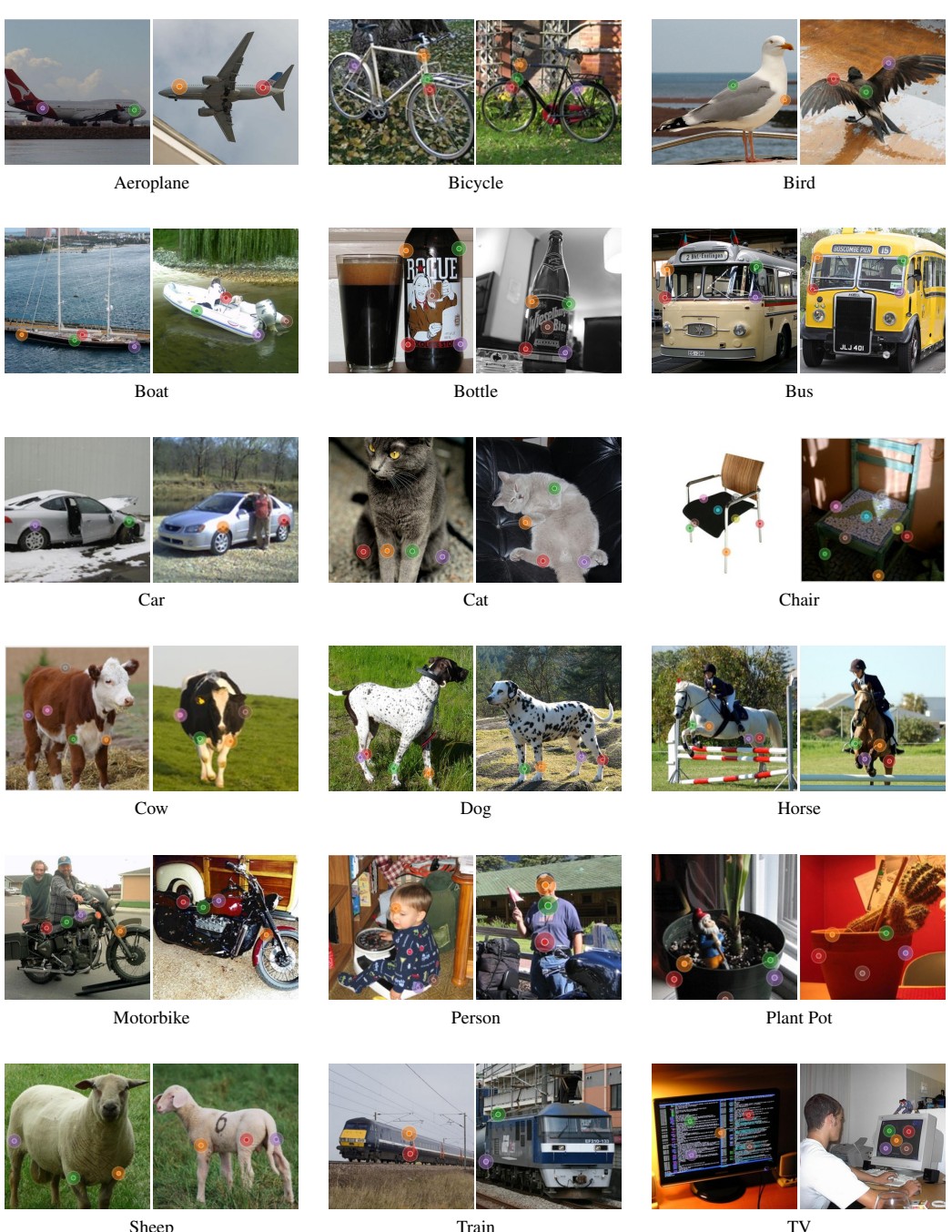

Figure A2: **Visualization of keypoint annotations from SPair-U.** Colors represent keypoint IDs.

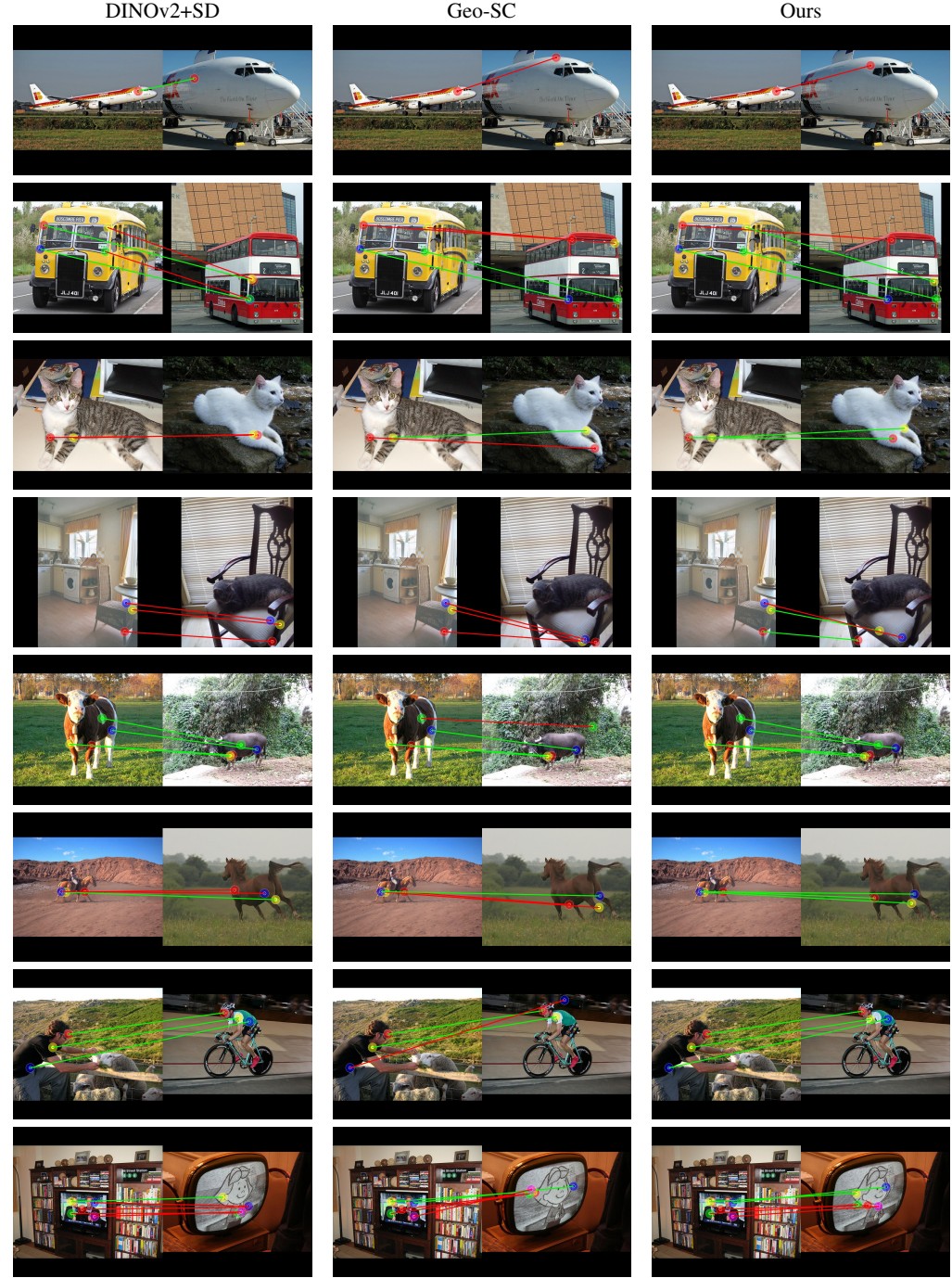

Figure A3: **Visualization matches for SPair-U.** Green lines are correct, red ones are incorrect.

DINOv2+SD                    Geo-SC                    **Ours**

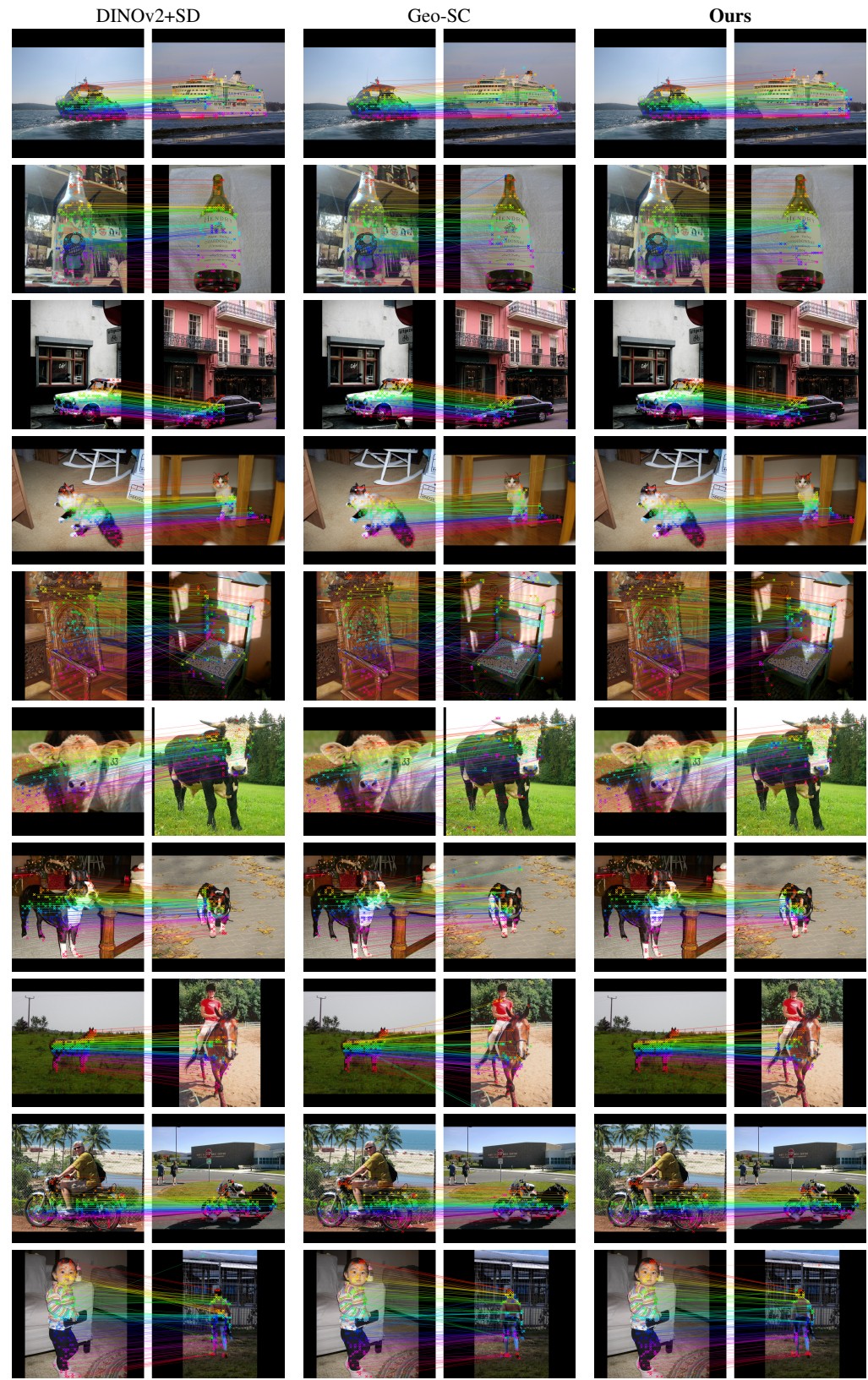

Figure A4: **Visualization matches for randomly selected object points.** Colors are provided as a way to distinguish the points.

