# OpenReview forum: "Jamais Vu: Exposing the Generalization Gap in Supervised Semantic Correspondence"
_NeurIPS.cc/2025/Conference — NeurIPS 2025 poster_

### Official Review · Reviewer_tgz1 · 2025-06-14

**Clarity:** 3
**Significance:** 2
**Originality:** 1
**Rating:** 4
**Confidence:** 5

**Summary:**

The paper proposes a semantic matching method, which establishes semantic correspondences between images by lifting the  2D keypoints into a canonical 3D space using monocular depth estimation.
The main motivation of the paper lies in the limited generalizability of existing semantic correspondence methods beyond the sparsely annotated training keypoints seen during training.
To better evaluate the generalizability of semantic matching methods on unseen keypoint categories, the paper further introduces introduce SPair-U, an extension of the existing SPair-71k benchmark with additional keypoint annotations.
The proposed method shows strong semantic matching performances across SPair-71k and SPair-U.

**Questions:**

Please refer to the weaknesses section. Specific questions below:

1. How does the proposed method differ from SemAlign3D, and what additional technical / conceptual novelties does this paper hold?

2. How would the proposed method perform without additional supervision (i.e., semantic meaning per keypoint), just using the keypoint positions as supervision?

3. How do other existing methods (especially semantic matching methods which focus on correlation map refinement) perform on SPair-71k and SPair-U?

**Ethical Concerns:**

["NO or VERY MINOR ethics concerns only"]

**Final Justification:**

The authors have cleared my major concerns during the rebuttal phase. While some questions remain regarding whether using semantically pre-defined keypoints can really handle the generalization gap, I hope the authors can handle this in the revision with additional analysis and visualizations.

**Limitations:**

Yes.

**Paper Formatting Concerns:**

No critical concerns regarding paper formatting.

**Quality:**

2

**Strengths And Weaknesses:**

### Strengths:

1. The paper addresses an important problem in semantic matching i.e., limited ability of existing methods in establishing dense semantic correspondences across viewpoints across the entire foreground of the objects.

2. The paper is well written and easy to follow.


### Weaknesses:

1. The paper shares strikingly similar idea with SemAlign3D, a paper cited by the authors in L111 (Related Work: Geometry-Aware Methods). The core idea of building per-class 3D canonical representations based on monocular depth maps and segmentation masks overlap significantly. The authors mention that SemAlign3D "requires segmentation masks and lacks the ability to generalize to new categories, as no prototype for them would exist for them (the new categories) in absence of training samples", the same limitations exist for the proposed method. Moreover, the reported performance on SPair-71k in this paper falls behind SemAlign3D. At the time of submission of this paper (mid May), SemAlign3D had been accepted to CVPR 2025 and was already uploaded on Arxiv (late March). The authors were aware of SemAlign3D -- with that in mind, it is challenging to notice additional technical contribution beyond SemAlign3D in this paper.

2. Disregarding the existence of SemAlign3D, the proposed pipeline is hard to agree with. Unlike existing semantic matching methods, the proposed method assumes that the object-wise keypoints are always on **semantically pre-defined positions**, and narrows the problem of semantic matching down to matching pre-defined semantic keypoints. The proposed method therefore aims to "produce distinctive and semantically consistent features across object instances for each keypoint k (L 168)", which, I believe, contradicts with the motivation of the paper which aims to address the "limited generalizability of existing methods". In clearer terms, I believe that assigning semantic meanings to each keypoint per object can be regarded as 'additional supervision' beyond simple keypoint positions, and this poses unfair comparisons with existing work.

3. The proposed method lacks comparison with other existing work, which may show strong performances on SPair-71k and SPair-U. For example, semantic matching methods which refines correlation maps e.g., [1][2], can be expected to perform better in dense semantic matching scenarios.

[1] Min et al., Convolutional Hough Matching Networks, 2021
[2] Kim et al., TransforMatcher: Match-to-Match Attention for Semantic Correspondence, 2022

---

> ### Author Rebuttal · Authors · 2025-07-30
>
> We thank **tgz1** for their valuable feedback and for acknowledging our work addresses an important limitation in existing methods. To summarize, the primary concern noted by **tgz1** relates to the perceived similarity between our approach and the concurrent work SemAlign3D (Wandel et al., CVPR 2025). In our response below we go into detail describing why (1) according to the official NeurIPS guidelines it should be considered concurrent work and thus does not need to be compared against and (2) both works, while sharing similar high-level motivation, are actually significantly different in terms of their approaches (e.g., SemAlign3D is a test-time optimization method with no learnable components). As a reminder, in addition to proposing a new approach for semantic correspondence estimation, our core contribution relates to identifying a deeply rooted issue in the way supervised semantic correspondence pipelines are trained and evaluated.
>
> **[tgz1-1] Comparison to SemAlign3D.**
> The official NeurIPS 2025 call for papers states that *“papers that appeared online after March 1st, 2025 will generally be considered ‘contemporaneous’ in the sense that the submission will not be rejected on the basis of the comparison to contemporaneous work.”*
>
> SemAlign3D was published at CVPR 2025 and first made public on arXiv on 28th of March 2025, which clearly identifies it as contemporaneous. We would like to clarify that we developed our work independently from SemAlign3D. We only became aware of it shortly before the NeurIPS deadline, we did our best to acknowledge it by citing it as concurrent work (L111) and submitted our work in good faith. Note, the code for SemAlign3D was only released on July 10th, but this is only a subset of the code needed to recreate the results in their paper. Below is a breakdown the main conceptual and technical differences between our works:
>
> *[tgz1-1.1] Paper contributions.*
> SemAlign3D proposes a new supervised semantic correspondence system and introduces a new approach based on matching objects through 3D category prototypes built by lifting a segmentation and monocular depth model. In contrast, our primary goal is to expose a critical flaw in recent semantic correspondence benchmarks and model designs. We do so by showing that recent supervised models are overfitted to the training set and merely act as keypoint detectors (see Table 1 vs 2). In order to demonstrate this, we introduce new evaluation points that do not appear in the SPair-71k training set. Finally, we hypothesize that incorporating 3D awareness is an efficient solution to this generalization gap, and introduce a model following this principle that shows better generalization on SPair-U to validate it. The only point of confluence is that both our model and SemAlign3D leverage 3D category prototypes. However, other prior works have also previously explored the use of 3D prototypes for correspondence learning, e.g., [27,39]. Please refer to L97-115 in our paper for a discussion of geometry aware methods.
>
> *[tgz1-1.2] Conceptual model differences and associated limitations.*
> The way SemAlign3D and our approach use the 3D prototypes is also conceptually different, incurring different limitations. SemAlign3D does not train any neural network. It builds the 3D prototypes using the training images, and aligns them at inference time as a test-time optimization procedure. As a result, their approach has several limitations:
> 1. To align the correct category prototype on the test image, it needs to have seen the exact object category during training, e.g. build a chair prototype to be able to evaluate correspondence on test chairs, preventing generalization to new categories.
> 2. It needs to know the object category at test time, to select the corresponding prototype.
> 3. The test-time optimization process relies on a **test** segmentation mask (Eq. 11 in SemAlign3D).
> 4. Finally, test-time optimization is costly, requiring 10-30 seconds to align the prototype on a single image, so up to a full minute for a pair (Sec. 4.3, paragraph 1 in SemAlign3D). It is also prone to failure and requires category-specific hyperparameters (Optimization paragraph in Sec. 4.1 and Supplementary Tab. 5 in SemAlign3D)
>
> In comparison, our model aims to learn a function $\phi$, parametrized by a neural network, that enables general 2D correspondence prediction. The 3D prototypes we create are only used during training to allow us to compute our geometric loss (Eq. 6). At inference time, features predicted by $\phi$ are directly used to compute correspondence, which means:
> 1. We can apply $\phi$ to images of any category and estimate correspondence on them. We do so in our cross-benchmark evaluation (L264) where we evaluate models trained on Spair-71k on other datasets, including AP-10K which contains animal categories very different from those seen during training, e.g., monkeys, bears and squirrels. Results in Table 3 show that our model generalizes better than its competitors in this challenge regime.
> 2. We train a single $\phi$ shared across all categories and thus do not require knowing the test category beforehand.
> 3. We compute correspondence without requiring the aid of a test-time segmentation mask.
> 4. Estimating correspondence between two images only requires computing two forward passes and querying nearest neighbors. On our hardware which is roughly similar to that used in SemAlign3D, it takes only 50 ms per pair, meaning our model is roughly **three orders of magnitude faster** than theirs (x400 up to x1200).
>
> *[tgz1-1.3] Technical model differences.*
> While both approaches rely on segmenting the object and using a monocular depth model to obtain a point cloud, the way we build our 3D prototypes is also different. Semalign3D relies on estimating the distribution of relative distance and angles of keypoint quadruplets, then explicitly learns a dense prototype by clustering points coming from different object point clouds. In contrast, we only learn a coarse category prototype using Kabsch’s algorithm. Our dense mapping is implicitly learned through geometrically aligning the predictions of  $\phi$ to with the backprojected point clouds. Thus, both 3D prototypes are learned in a completely different way, and only share some high-level conceptual similarity.
>
> We hope this clearly shows that our paper shares a similar motivation in its model design with SemAlign3D simply because the idea of developing a 3D category prototype is an intuitive solution to solving correspondence. We are happy to discuss these points further during the discussion period should any clarification be needed. We will revise our discussion of SemAlign3D, which is indeed very interesting work, in the updated paper to summarize these key differences.
>
> **[tgz1-2] Keypoint semantics as extra supervision?**
> Irrespective of how models were trained, we *precisely* show that previous models overfit to semantically predefined positions, as they are unable to generalize to new semantic points that do not appear in the training annotations (see Fig. 1). Even if such an assumption is not explicitly made in their design, they still end up only being able to match predefined keypoints due to the limitation of the task design, i.e., sparse supervision and semantic overlap between the training and evaluation points, **which is a central point of our paper**.
>
> Moreover, gathering all points that belong to the same semantic part, even if the keypoint semantics are not provided, can easily be done by iterating through the training images and grouping points by equivalence class, e.g., point A in image 1 corresponds to point B in image 2 and point C in image 3, etc…, hence all these points share the same semantics. Therefore, we respectfully disagree that this constitutes ‘additional supervision’ as we use exactly the same labels as other models, albeit under slightly different assumptions.
>
> In practice, all semantic correspondence datasets are made up of instances labeled with a predefined keypoint set. That is because semantics are by definition human-defined, and we have a bias towards salient points common to all instances, therefore expecting anything else is unrealistic and would constitute a particularly contrived setting.
>
> Finally, similar or even more constraining assumptions are made in recent related works. As previously discussed, SemAlign3D makes the same assumptions as us, while Geo-SC explicitly requires additional semantic knowledge to perform its flipping based data augmentation (when flipping the image of a person, a left hand becomes a right hand).
>
> **[tgz1-3] Comparison to CHMN and TransfoMatcher.**
> Thanks for pointing out these papers, we will add them to our related works section. While their dense objectives could indeed make them less prone to overfitting, their designs lack 3D awareness, which we posit is an important component for generalization, especially under the large viewpoint changes that SPair exhibits.
> We evaluated both approaches on SPair-71k and SPair-U. For CHMN, we used the code and checkpoint provided by the official github repo. For TransforMatcher, we used the code provided by the official github repo. Since no checkpoint was released, we retrained the model ourselves.
>
> | PCK@0.1$_{bbox}$ | SPair-71k| SPair-U|
> |-|-:|-:|
> |CHMN| 46.4 | 28.0|
> |TransfoMatcher| 50.7 | 29.7|
> |ScorrSan|55.3|32.7|
> |Ours| 85.4| 66.1|
>
> Both models exhibit a significant generalization gap, very consistent with that of the other related works. The closest performing model we evaluated in the paper, SCorrSan, reaches similar numbers. Interestingly, SCorrSan also aims to learn dense descriptors, though it uses pseudo labels to achieve it. Generally speaking, this confirms our observation that many current methods overfit to the seen keypoints in SPair-71k and fail to generalize well to unseen ones. We will add these news results to the revised paper.

---

> > ### Comment · Reviewer_tgz1 · 2025-08-05
> >
> > I thank the authors for their detailed clarification in their comparison to SemAlign3D. I apologize for the confusion -- just to clarify, I was not trying to point to SemAlign3D as grounds for rejection, but I was wondering why the authors were mentioning SemAlign3D and highlighting their weaknesses when I could not notice strong differences at the time of my review. Thanks to the rebuttal, I have a much clearer understanding on their differences.
> >
> > However, while the authors propose that "previous models overfit to semantically predefined positions, as they are unable to generalize to new semantic points that do not appear in the training annotations", I think visualizing more correspondences from Jamais Vu would be effective. For example, in Figure 15 of "Convolutional Hough Matching Networks for Robust and Efficient Visual Correspondence", it can be seen that correspondences are being made across non-predefined positions, nearly akin to co-segmentation. My main concern lies in whether the proposed method 'really' can adapt to non-predefined positions, beyond the keypoints in Spair-U.
> >
> > However, the remaining concern is relatively minor; having read through the other reviews as well, I am raising my rating. I thank the authors once again for their clear and detailed rebuttal.

---

> > > ### Author Response · Authors · 2025-08-06
> > >
> > > Thanks for carefully considering our rebuttal. As stated in our SPair-U performance analysis (L260-263), we agree that our model does not fully adapt to non-predefined positions: it showcases a substantial generalization gap, although significantly less so than related approaches. We believe the core contribution of our work is illustrating this gap to allow future research to have a critical eye on the matter when developing new models and benchmarks. The visualization of additional matches would indeed be a very clear way to show the generalization ability of models to new points. We will add some alongside the SPair-U matches already present in the supplementary (Fig. A3).

---

### Official Review · Reviewer_UJnP · 2025-06-20

**Clarity:** 3
**Significance:** 2
**Originality:** 2
**Rating:** 3
**Confidence:** 4

**Summary:**

This paper proposes a method for constructing a canonical 3D representation for object categories and using it to establish semantic correspondences.

**Questions:**

Please address the concerns expressed in the weaknesses part of the review above.

Additional questions:

As far as I know, Kabsch's algorithm (or its alternatives using quaternions) find a rigid transformation, not a similarity, please clarify.

It appears to be the case that the number of key points is fixed over all category instances. For a true 3D object see from different viewpoints, half of the object (its back) will always be invisible. How do you expect to handle this problem?

I didn't understand your geodesic sampling strategy : what is the geodesic distance you use since you don't know the surface of the object or the true connectivity of the points? Please clarify.

**Ethical Concerns:**

["NO or VERY MINOR ethics concerns only"]

**Final Justification:**

I appreciate the efforts the authors have put in the rebuttal, but in the end my main concern, the implicit assumption that objects within a category have the same shape, stands, and I thus also stand with my rating.

**Limitations:**

Yes

**Paper Formatting Concerns:**

No concerns

**Quality:**

3

**Strengths And Weaknesses:**

Strengths:
The paper is well written, and the proposed method gives good results, in particular on data for which some keypoints are available for testing, but were not annotated during training.

Weaknesses:
The presentation appears to implicitly assume that all instances of the same object category have the same 3D shape since Eq. (3) expresses the fact that their keypoints are related by similarity transforms. This is a very strong assumption that will not hold for many object categories (think ot cats or even cars for example), and a major limitation of the proposed approach. The necessity to relax global rigidity in  geometric representations of object categories was already noted in old papers such as Lazebnik et al., "Semi-local affine parts for object recognition", BMVC'04 for example.

In Eq. (3) there does not appear to be any constraint on the points p_i and matrices M. But  in this case what prevents the optimization from reaching a global optimum of 0 with p_i=0 and M=0?

The proposed method learns two descriptors z and Phi, one for the canonical keypoints, and one of image keypoints. But there does not appear to be any constraint on these either, except they should be similar. What makes these descriptors informative, and what prevents them from collapsing into constant values for example?

---

> ### Author Rebuttal · Authors · 2025-07-30
>
> We thank **UJnP** for their insightful comments, and recognizing the ability of our model to generalize to points that weren’t seen during training. As an initial remark, we would like to emphasize that our paper goes beyond simply proposing a new semantic correspondence approach and new benchmark dataset. Our goal is to expose a fundamental flaw in the way supervised semantic correspondence pipelines are trained and evaluated, i.e. **issues in the general task design** are the reason behind the generalization gap observed in models. We believe this to be our work’s core contribution, of particular significance to the community as it carries strong implications towards future model and benchmark design. In particular, GeoSC - the top performing model on SPair-71k at the time we wrote our paper - trains a supervised feature predictor on top of frozen DINOv2 and SD features, yet when evaluated on SPair-U it performs *worse* than simply using the DINOv2 and SD features (see Tables 1 and 2). This means that **none** of what it has learned transfers to new semantics, effectively failing to solve anything beyond sparse keypoint detection.  While the raised concerns about restricted model design and missing technical details are valid, we are concerned that this contribution may have been overlooked and would like to highlight it.
>
> **[UJnP-1] Assuming all instances of a category have the same shape?**
> We agree that making a global rigidity assumption in Eq. 3 is a very coarse approximation, and can appear to contradict our goal of learning a correspondence model for deformable objects. However, this **global** assumption is only made in Eq. 3, whose sole purpose is to optimize the set of 3D keypoints $\mathcal{P}$ into a coarse spatial organisation of 3D keypoints, e.g., making sure that the left hand keypoint generally sits opposite of the right one, in order to provide a reasonable structure for our canonical prototypes. Importantly, $\mathcal{P}$ is required to compute our geometric losses, *but is not used to predict correspondence*.
>
> The component of our model that predicts correspondence is the feature predictor $\phi$, which is not directly involved in Eq. 3. It is trained through Eq. 5 and 6 in which only a **local** rigidity assumption is made. Though they happen simultaneously, optimizing $\mathcal{P}$ and $\phi$ can effectively be considered separate processes as $\mathcal{P}$ is considered fixed in Eq. 5 and 6.
>
> Visualizations of the learned $\mathcal{P}$ (the large points in Fig 5) illustrate that the assumption of global rigidity *for $\mathcal{P}$ only* allows our model to recover a plausible 3D structure, even if in practice the estimated optimal rigid alignment $\hat{\mathbf{M}}^{(n)}$  is far from perfect and still incurs relatively large errors due to object deformations.
>
> **[UJnP-2] Missing constraint in Eq. 3?**
> Apologies for this missing detail. In practice the estimated scale $s$ is constrained to be larger than 1, i.e., instances can only be scaled up, not down. This has the effect of rescaling all instances to the size of the largest one in the training set, and prevents collapse. We will update the text to make this clear.
>
> **[UJnP-3] Missing constraint between $z$ and $\phi$?**
> As mentioned in Sec 5.1, we build our model on top of Geo-SC [51]. The main training objective of Geo-SC is a CLIP-style contrastive loss between source and target keypoints. This prevents collapse of the descriptors predicted by $\phi$, which in turn prevents collapse of $\mathcal{Z}$ through Eq.4. We will also update the text to make this clear.
>
> **[UJnP-4] Kabsch algorithm.**
> You are correct, this is a misuse of the term on our part. To be precise, estimating $M$ requires solving a *constrained orthogonal Procrustes problem*. Only $R$, the rotational part of $M$, is obtained through the Kabsch algorithm.
>
> **[UJnP-5] Occluded points.**
> For each training image, we use all visible keypoints, and only these points, i.e., the summations in Eq.3 are in fact not computed over all points in $\mathcal{P}$, only those that correspond to the annotated points of $I$. At inference time, correspondences are computed using $\phi$ only, i.e., $\mathcal{P}$ is not involved.
>
> **[UJnP-6] Geodesic sampling.**
> We named this procedure “geodesic sampling" as its goal is to aggregate neighbors along the object surface. However, it does not explicitly use the geodesic distance as we do not have access to the connectivity of the points. Below is a pseudocode for it, which we will add to the implementation details:
>
> Input: Point cloud $\mathcal{X}$, seed point $p$, number of neighbors $k$;
> Output: neighbor set $B$;
> $B \leftarrow$ {$p$}
> while $|B| < k$:
> |&nbsp;$\mathcal{X}' \leftarrow \mathcal{X}\setminus B$   # We do not consider already selected points
> |&nbsp;for $x$ in $\mathcal{X}$':
> |&nbsp;|&nbsp;$D_x \leftarrow \min_{y\in B} \left \lVert x - y \right \rVert_{2}$ # Compute the distance to the closest point in B
> |&nbsp;$B \leftarrow B \cup$ {$\arg\min_{x\in \mathcal{X}'} D_x$} # Add the point with minimal distance to B
> return $B$
>
> By iteratively adding points that minimize the distance to the set of already selected neighbors, we minimize the risk of jumping over surface discontinuities.
>
> **[UJnP-7] Additional related work.**
> Thanks for pointing out Lazebnik et al. BMVC 2004. Their work is concerned with finding geometrically invariant parts for modeling 3D objects in the context of object recognition, and not our explored task of semantic keypoint correspondence. While their goal of matching image regions using local affine transform bears some high-level similarities with our local geometric objective defined by Eq. 5 and 6, ours is performed in 3D, endowing our model with the ability to deal with large viewpoint changes.  We will add a discussion to our related work.

---

> > ### Comment · Reviewer_UJnP · 2025-08-02
> >
> > Thank you for your response and for clarifying some issues, for example defining your geodesic sampling strategy. My main concern, however, still remains: the proposed representation assumes through Eq. (3) that all instances of the same category have the same 3D shape. Even if this is the only place this assumption is used, it is a key to the construction of the canonical keypoints used throughout.
> > For Klabsch's algorithm, since it only finds the rotational part of the similarity function, how do you find the scale?
> > Finally the reference to Lazebnik et al. was not intended to be there as a reference to work similar to your, but just as an example that there are well known ways to relax rigidity constraints when dealing with geometric representations of objet categories.

---

> > > ### Author Response · Authors · 2025-08-04
> > >
> > > We thank **UJnP** for reading our rebuttal and for following up. We appreciate the opportunity to further clarify questions around our learned per-category prototypes and their construction.
> > >
> > > **Do all instances of the same category have the same 3D shape?**
> > > We respectfully disagree that our representation $\phi$ assumes that all instances of a category have the same rigid 3D shape. *Our core assumption is that all instances of a category can be smoothly mapped to the same canonical shape.* $\mathcal{P}$ is only used as a parametrization of the canonical shape used to supervise $\phi$, and many such parametrizations are valid. For instance, a canonical human could be represented via a T-pose, or by having its arms pointing forwards, upwards, or downwards, all of which would allow the computation of Eqs. 5 and 6. The key insight is that once we have the canonical shape for a category, we associate a set of learnable descriptors to it (see Eq. 4) which are then used to match keypoints from individual object instances to the canonical shape.
> > >
> > > Eq. 3 simply provides a practical way to estimate a reasonable $\mathcal{P}$, i.e., one that does not cause extreme deformations or self-intersections. It is **not** fundamental to the model, e.g., any procedure that delivers a reasonable $\mathcal{P}$ can replace it. In the new ablation study (see [Avym-2]) we substitute a local-rigidity variant that aligns only triplets of neighbouring keypoints, which would mean making only a local rigidity assumption, not dissimilar to Lazebnik et al. BMVC 2004. The results yield virtually identical performance to the full global-rigidity version, confirming that our approach does not rely on the assumption that all instances share one rigid 3D shape.
> > >
> > >
> > > Finally, we believe that our experiments clearly demonstrate that recovering $\mathcal{P}$ with a global alignment is not a limitation even for deformable objects like cats, cars, and humans. Specifically on SPair-U, our model shows significantly improved generalization performance compared with its backbone Geo-SC on birds (+16.5 pts), cats (+11.1 pts), and humans (+7.0 pts). Qualitatively, we show plausible learned canonical shapes in Fig. 5, and the feature maps showing clear separation of individual limbs on the animals and the person in Fig. 4 illustrates our ability to deal with deformations. We would gladly welcome any examples in which our assumptions might lead to issues so we can address them thoroughly.
> > >
> > > **Obtaining the scale.**
> > > After centering the points clouds to obtain $\mathbf{X = \\{x_i\\}}$ and $\mathbf{Y = \\{y_i\\}}$ and computing the optimal rotation $\mathbf{R}$ between them, we obtain the optimal scale by minimizing the squared error $E$ between the two sets according to a global scaling $s$, i.e.,  $s^* = \min_s \sum_i ||\mathbf{y_i} - s\mathbf{Rx_i}||_2^2$. This can be computed by solving $\frac{\partial E}{\partial s} = 0$, leading to the closed form solution $s^* = \sum_i \mathbf{y_i}^T\mathbf{Rx_i} / \sum_i ||\mathbf{x_i}||_2^2$. This is also known as the Umeyama procedure [1], and an implementation of it which served as the basis for ours can be found in PyTorch3D’s `corresponding_points_alignment` function.
> > >
> > > **RE Lazebnik et al.**
> > > Thanks for clarifying.
> > >
> > > [1] Umeyama et al. "Least-squares estimation of transformation parameters between two point patterns.", PAMI, 1991

---

> > > > ### Comment · Reviewer_UJnP · 2025-08-05
> > > >
> > > > Thank you for your response and clarifying your alignment procedure (although it does not account for translation, I trust you that it is correct). I have read your response and the response Avym-2, and using fewer points than those visible does not make the model locally rigid, it just uses fewer points to enforce rigidity. The fact remains that the construction of the canonical  shape that is at the core of your model does implicitly assume a rigid body. The fact that you obtain good results is commendable but does not change this fact and I will keep my rating.

---

> > > > > ### Author Response · Authors · 2025-08-06
> > > > >
> > > > > Thanks for following up - we appreciate your input. We agree that the learned canonical space for each category is represented by a single rigid prototype. However, our experimental evaluation indicates that this assumption does not impact our ability to estimate correspondence across highly deformable categories. This is due to how we match instances to the prototype, which does not make such a rigid assumption. If there are suggestions for categories or examples where this assumption is actually an issue we are happy to include them as limitations in the revised text.

---

### Official Review · Reviewer_Avym · 2025-06-23

**Clarity:** 2
**Significance:** 2
**Originality:** 2
**Rating:** 4
**Confidence:** 4

**Summary:**

Based on the observation that current supervised methods for category-specific 2D semantic correspondence fail to generalize to unseen keypoints not annotated during training, the authors proposed a new approach using geometric constraints at training time through a per-category, learned continuous canonical manifold.

**Questions:**

1. In the beginning of section 3.1, the authors introduced a notation S, which is never referred to in the following text. Also, from the whole section 3, it’s unclear whether the method wants to optimize dense feature descriptors for dense 3d points on the continuous canonical manifold or just optimizing feature descriptors for the |K| number of 3d keypoints.
2. In the equation (3) in section 3.2, it’s unclear how to do the optimization, since optimizing M requires p, and optimizing p requires M. I guess the optimization is conducted alternatively, but it’s unclear in the text.
3. For the equation (3) in section 3.2, the authors want to optimize a category-specific sparse 3d keypoint set P for all input. But objects within input images will have different articulation and arbitrary deformations, which will lead to \bar{k} got by equation (2) from different images cannot be described accurately by only rigid transformation (with scaling) of just one 3d keypoint set P. If the authors’ idea is to only approximately learn a roughly correct rigid transformation (with scaling) M and a roughly correct 3d keypoint set P from equation (3), then ablation studies on different selection of 2d keypoints annotations need to be done to validate the method.
4. In section 3.3, equation (6) defined an optimization based on local small neighbors of canonical space C and posed space B, but the details of how to select the neighbor size k and the ablation study of how the size k will influence the results (especially for different categories, since objects within difference categories will deform in difference sense) are not included in the paper.

**Ethical Concerns:**

["NO or VERY MINOR ethics concerns only"]

**Final Justification:**

After reading the rebuttal, my most concerns are well addressed. Therefore, I would like to raise my final score.

**Limitations:**

1. The writing quality of the paper needs to be improved, since there are some technical details are unclear as mentioned in the questions section.
2. The proposed method relies on the off-the-shelf segmentation and depth estimation methods and thus the performance heavily dependent on the quality as well as the consistency of the segmentation and depth estimation results. It is expected to evaluate how robust of the proposed method for highly articulated object such as human, animal, where the segmentation and depth estimation is typically less accurate due to the large deformation as well as occlusion.

**Paper Formatting Concerns:**

No paper formatting concerns.

**Quality:**

2

**Strengths And Weaknesses:**

1. The authors proposed a method that leveraged pre-trained depth predictor to give depth estimation for sparse annotated 2d keypoints in image to lift them to 3d, and optimize a canonical 3d keypoints set as correspondence reference set. And the extra introduced 3d information will make 2d correspondence be more accurate.
2. The authors also proposed a method that generate dense 3d keypoints from the optimized sparse 3d keypoints set, will make the optimized semantic correspondence prediction model could generalize to new 2d labelled keypoints unseen during training.
3. The authors proposed a new dataset SPair-U, which will benefit future researchers in 2d semantic correspondence area.

---

> ### Author Rebuttal · Authors · 2025-07-30
>
> We thank **Avym** for their detailed comments, and endorsing the use of monocular depth models to learn a dense generalization of sparse keypoint, and the benefits of introducing SPair-U. First and foremost, we would like to emphasize that our paper goes beyond simply proposing a new semantic correspondence approach and new benchmark dataset. Our goal is to expose a fundamental flaw in the way supervised semantic correspondence pipelines are trained and evaluated, i.e. **issues in the general task design** are the reason behind the generalization gap observed in models. We believe this to be our work’s core contribution, of particular significance to the community as it carries strong implications towards future model and benchmark design. In particular, GeoSC - the top performing model on SPair-71k at the time we wrote our paper - trains a supervised feature predictor on top of frozen DINOv2 and SD features, yet when evaluated on SPair-U it performs *worse* than simply using the DINOv2 and SD features (see Tables 1 and 2). This means that **none** of what it has learned transfers to new semantics, effectively failing to solve anything beyond sparse keypoint detection. While the concerns about unclear technical details and missing ablations are valid, we are concerned that this contribution may have been overlooked and would like to highlight it.
>
>
> **[Avym-1] Unclear technical details.**
> Thank you for pointing these out, we will update our manuscript to clarify these points
> * We will remove the explicit definition of $S$.
> * Our goal is to learn a dense feature predictor $\phi$ that effectively predicts semantic correspondence. We do so by leveraging the sparse $|K|$ annotated keypoints to coarsely align all instances of an object category, then rely on fine-grained geometric constraints to densely supervise $\phi$.
> * In Eq. 3, only $\mathcal{P}$ is optimized. $\hat{\mathbf{M}}^{(n)}$ is not optimized, it is obtained analytically by running an orthogonal procrustes solver relying on the Kabsch algorithm. The mention of a “nested optimization” on L159 was a misnomer on our part.
>
> **[Avym-2] Learning the roughly correct rigid transformation.**
> The point of Eq. 3 is to guide $\mathcal{P}$ towards a coarse category shape to serve as a reasonable basis for our canonical dense mapping. Because $\hat{\mathbf{M}}^{(n)}$ is rigid and objects can deform, it will indeed only be “roughly correct”. A detailed explanation of this point can be found in response [fqdv-3].  For each training image, *we use all visible keypoints*, and only these points, i.e., the summations in Eq. 3 are in fact not computed over all points in $\mathcal{P}$, only those that correspond to the annotated points of $I$. As requested, we perform ablations by randomly selecting a subset of points when computing Eq. 3. Selecting fewer than 3 would lead to a degenerate alignment problem, and there are few image pairs in which more than 8 are visible at the same time.
>
> Here are the different PCK@0.1$_{bbox}$ on the SPair-71k *validation* set.
>
> | #selected points|     3|      4|      5|     6|    7|      8|*all*|
> |:----------------------|-----:|------:|-----:|-----:|-----:|-----:|-----:|
> | PCK                  | 86.6| 86.4|86.1|86.4|86.5|86.0|86.5|
>
> Selecting a subset of points instead of using all of them only appears to cause small fluctuations in validation performance.
>
>
> **[Avym-3] Ablation of neighborhood size.**
> After performing backprojection, we first subsample 1024 points from the resulting point cloud using farthest point sampling in order to have a similar number of points per object instance. Then, for each of these points, we compute its 64 nearest neighbors to obtain $C_q$ . We selected a neighborhood size of 64 as it makes up 1/16 of the point cloud size, striking a good balance between being representative of the geometry (too few points carry little information about the object structure) while allowing deformations (too many and the rigidity assumption becomes global). As requested, we evaluated the impact of different neighborhood sizes, where we report PCK@0.1$_{bbox}$ on the SPair-71k *validation* set.
>
> | #NN|     4|      8|   16|   32|*64*|128| 256|
> |:------|-----:|------:|-----:|-----:|-----:|-----:|-----:|
> |PCK | 85.5| 86.1|85.9|86.5|86.5|86.6|86.1|
>
> Except for extreme values, the neighborhood size appears to have little impact on the validation performance on the seen keypoints in SPair-71k. We will add these additional results to the revised paper.
>
>
>
>
>
> **[Avym-4] Requires segmentation and depth which may be inaccurate.**
> Using the depth or segmentation predictions from pre-trained models to build 3D-aware representations has been adopted before in related works [3, 27, 42, 46]. In particular, concurrent work [46] uses both, and unlike us it even requires segmentation masks at *test-time*. In contrast, our approach only uses them during training. At inference time, correspondences are estimated solely using image features predicted by $\phi$. It would be possible to filter out training instances where segmentation or depth is poorly estimated before training to remove any harmful impact of them. In practice however, we did not observe any catastrophic failures of these predictions.
>
>
> **[Avym-5] Results on highly articulated objects such as human and animal.**
> Our model is evaluated on many deformable categories like people and animals from the SPair and AP-10K datasets, and performs well on them (Table 3, 4, and 5). Specifically on SPair-U, our model shows significantly improved generalization performance compared with its backbone Geo-SC on birds (+16.5 pts), cats (+11.1 pts) and humans (+7.0 pts).

---

> > ### Comment · Reviewer_Avym · 2025-08-05
> >
> > The rebuttal answers my questions and addresses my concerns. Therefore, I would like to raise my final rating.

---

### Official Review · Reviewer_fqdv · 2025-06-28

**Clarity:** 3
**Significance:** 3
**Originality:** 3
**Rating:** 5
**Confidence:** 4

**Summary:**

The authors propose a new method for semantic correspondence estimation. They first analyse prior supervised methods and find that they act like keypoint detectors by overfitting on the few annotated keypoint classes and thus do not generalise well.
In their approach, they propose to use pre-trained monocular depth estimation and semantic segmentation models to estimate the 3D geometry of object instances.
Using these point maps, they propose to learn a canonical object space per-category. They do so by finding the optimal rigid alignment between the sparse 3D point clouds stemming from the annotated keypoints in all training images.
Further, they optimise a latent embedding for each of these canonicalised 3D keypoints. Using the feature similarity of other pixels to these points, they obtain their respective location in the canonicalised 3D space by linear interpolation.
The local neighbourhoods of given points in the so-found 3D canonicalised point maps are then compared with the posed point clouds retrieved through the frozen depth estimation model as an additional loss.
The paper further introduces new labels to the SPair-71k dataset that are used to evaluate the generalisation against prior supervised methods.

**Questions:**

Just to make sure that this is not the case, can you confirm that the training/evaluation is not done per-category but over all classes at the same time? So, while category-specific P and Z are learned, the feature refinement is shared over all classes.

It would be good to show the performance of the proposed objectives without the Geo-SC losses, as the proposed losses should be sufficient for learning semantic correspondence, as mentioned by the authors.

Is the loss in Eq.6 computed for all points q that lie within the object mask? What is the neighbourhood size that is used to compute the loss?

Is gradient stop applied for the computation of Eq.6 wrt. the locations of canonicalised keypoints P or are they not only optimised via Eq.3 and also indirectly through the minimisation of Eq.6 (given the computation in Eq.5)?

**Ethical Concerns:**

["NO or VERY MINOR ethics concerns only"]

**Final Justification:**

After reading the rebuttal by the authors, which addressed my questions and concerns, I remain positive about the impact and contributions of the paper. While the global rigidity assumption is fairly strong for inherently non-rigid objects (e.g. cats), it seems to be inevitable to make some sort of assumption about the global distribution of keypoints. The paper demonstrates that this assumption helps closing the generalization gap of supervised semantic correspondence models and proposes an additional set of keypoint annotations to measure this generalization on the SPair dataset, which, when made public, will help advance the field further and make other methods more robust as well.

**Limitations:**

yes

**Quality:**

3

**Strengths And Weaknesses:**

**Strengths:**

The paper generally follows a convincing story, where the motivation of overfitted supervised semantic correspondence models is clearly motivated (Fig. 1).

The related work section is generally also well written, but missing a few important references (see weaknesses).

The authors propose a method that almost maintains the "overfitted" quality of supervised SC methods but enable improved correspondence for other points through their 3D-aware loss, while not requiring any additional supervision besides the annotated 2D keypoints and the class labels through the use of pre-trained 2D models.

Introducing a new test dataset for further evaluation on the SPair-71k dataset is helpful for the community.

**Weaknesses:**

The related work section includes references to some unsupervised semantic correspondence methods.
First, the statement of these methods being "completely unsupervised" (l.87) is possibly not true for all of the referenced papers as class labels are still required as weak supervision signal for category-specific learning.
Further, it would be advisable to include more works doing weakly supervised correspondence detection, such as (P)WarpC [1,2] or also include methods that perform 3D shape matching (which is essentially semantic matching) by using features extracted with foundational vision models [3,4,5].

Several details on the method/implementation are missing or not stated clearly, see the questions below for details.

The method is based on the assumption that annotated keypoints are geometrically salient (as all points must lie within the convex hull of the annotated keypoints), i.e., close to farthest points on the objects.
While this is often true, e.g. in SPair, assuming this might reduce generality to other datasets.

The authors state that their local neighbourhood rigidity loss (Eq. 6) is only computed for local neighbourhoods because of the difference between posed and canonical object shapes (l.180).
However, the alignment of posed annotated keypoints with the canonical coordinates in P (Eq.3) is based on exactly this assumption, where rigid transformations (+ isotropic scaling) are computed as transformations between posed and canonical objects.
It is unclear why this rigidity assumption should be valid for the alignment of sparse keypoints but not for dense points on the surface.

The paper is lacking important ablation experiments. The authors argue that their introduced "objectives suffice to learn a SC model" but only show the results of the method in combination with the objectives from Geo-SC.
While leaving out single loss terms of the proposed new losses might not be possible (l. 24, suppl.), the results of this experiment should have been shown.

---

[1] Truong, Prune, et al. "Warp consistency for unsupervised learning of dense correspondences." Proceedings of the IEEE/CVF international conference on computer vision. 2021.

[2] Truong, Prune, et al. "Probabilistic warp consistency for weakly-supervised semantic correspondences." Proceedings of the IEEE/CVF Conference on Computer Vision and Pattern Recognition. 2022.

[3] Dutt, Niladri Shekhar, Sanjeev Muralikrishnan, and Niloy J. Mitra. "Diffusion 3d features (diff3f): Decorating untextured shapes with distilled semantic features." Proceedings of the IEEE/CVF Conference on Computer Vision and Pattern Recognition. 2024.

[4] Wimmer, Thomas, Peter Wonka, and Maks Ovsjanikov. "Back to 3d: Few-shot 3d keypoint detection with back-projected 2d features." Proceedings of the IEEE/CVF Conference on Computer Vision and Pattern Recognition. 2024.

[5] Uzolas, Lukas, Elmar Eisemann, and Petr Kellnhofer. "Surface-Aware Distilled 3D Semantic Features." arXiv preprint arXiv:2503.18254 (2025).

---

> ### Author Rebuttal · Authors · 2025-07-30
>
> We thank **fqdv** for their constructive review and for highlighting our clear motivation, coherent experimental narrative, and the value of SPair-U for testing generalisation. As an initial remark, we would like to emphasize that our paper goes beyond simply proposing a new semantic correspondence approach and new benchmark dataset. Our goal is to expose a fundamental flaw in the way supervised semantic correspondence pipelines are trained and evaluated, i.e. **issues in the general task design** are the reason behind the generalization gap observed in models. We believe this to be our work’s core contribution, of particular significance to the community as it carries strong implications towards future model and benchmark design. In particular, GeoSC - the top performing model on SPair-71k at the time we wrote our paper - trains a supervised feature predictor on top of frozen DINOv2 and SD features, yet when evaluated on SPair-U it performs *worse* than simply using the DINOv2 and SD features (see Tables 1 and 2). This means that **none** of what it has learned transfers to new semantics, effectively failing to solve anything beyond sparse keypoint detection. While the raised concerns about model limitations are valid, we are concerned that this contribution may have been overlooked and would like to highlight it.
>
> **[fqdv-1] Additional weakly supervised approaches.**
> We will update the related works section accordingly and clarify that [1,50] do not make any assumption about the category, while [2,38,39] typically train and evaluate their models on single category datasets.  Warp-based matching is indeed an important precursor of geometry-aware methods, however, it is largely limited to instances in the same pose. While our model design is performing some form of 3D shape matching, we work only with images and do not require meshes, relying instead on pretrained segmentation and depth to obtain partial point clouds, which is a weaker supervision signal.
>
> **[fqdv-2] Assume that annotated keypoints are geometrically salient.**
> While we agree it is a limitation (as discussed in section 6), to the best of our knowledge, it is unlikely to be a strong issue as semantics are by definition human-defined, and we have a bias towards geometrically salient points. In particular, it is unlikely that human annotators will not label object extremities, providing a good approximation of the convex hull.
>
> **[fqdv-3] Rigidity assumption.**
> We agree that making a global rigidity assumption in Eq.3 is a very coarse approximation, and can appear to contradict our goal of learning a correspondence model for deformable objects. However, this **global** assumption is only made in Eq.3, whose sole purpose is to optimize the set of 3D keypoints $\mathcal{P}$ into a coarse spatial organisation of 3D keypoints, e.g., making sure that the left hand keypoint generally sits opposite of the right one, in order to provide a reasonable structure for our canonical prototypes. Importantly, $\mathcal{P}$ is required to compute our geometric losses, *but is not used to predict correspondence*.
>
> The component of our model that predicts correspondence is the feature predictor $\phi$, which is not directly involved in Eq.3. It is trained through Eq.5 and 6 in which only a **local** rigidity assumption is made. Though they happen simultaneously, optimizing $\mathcal{P}$ and $\phi$ can effectively be considered separate processes as $\mathcal{P}$ is considered fixed in Eq.5 and 6.
>
> Visualizations of the learned $\mathcal{P}$ (the large points in Fig 5) illustrate that the assumption of global rigidity *for $\mathcal{P}$ only* allows our model to recover a reasonable 3D structure, even if in practice the estimated optimal rigid alignment $\hat{\mathbf{M}}^{(n)}$  is far from perfect and still incurs relatively large errors due to object deformations.
>
>
> **[fqdv-4] Ablation without GeoSC.**
> Thank you for pointing it out, it is indeed an important ablation that we overlooked. We trained a model purely from DINOv2 and SD features, disabling all components introduced in GeoSC. It obtained a PCK@0.1 of 75.9 on SPair-71k, 0.6 points lower than the simple supervised baseline trained only on these features. On SPair-U, it reaches a PCK of 66.0, only 0.1 points below what our complete model achieves. This demonstrates surprisingly well that what GeoSC learns and what our geometric losses learn are in fact almost orthogonal, and strengthens our observation that GeoSC fully overfits the training semantic, as removing it from our model does not harm the performance on SPair-U.
>
> **[fqdv-5] A single model for all categories.**
> Yes, as noted at  L267 we train our model on all categories jointly. Each category has its own $\mathcal{P}$ and $\mathcal{Z}$, while $\phi$ is the same for all, allowing generalization to new categories as illustrated in Table 3. We will update the revised text to make this clearer.
>
> **[fqdv- 6] Additional implementation details.**
> After performing backprojection, we first subsample 1024 points from the resulting point cloud using farthest point sampling in order to have a similar number of points per object instance. Then, for each of these points, we compute its 64 nearest neighbors to obtain $\mathbf{C}_q$ . We selected a neighborhood size of 64 as it makes up 1/16 of the point cloud size, striking a good balance between being representative of the geometry (too few points carry little information about the object structure) while allowing deformations (too many and the rigidity assumption becomes global). We perform an ablation study of this in our response [Avym-3].
>
> Yes, $\mathcal{P}$ is only optimized using Eq. 3. They are considered fixed in Eq. 5 and 6 and no gradient is propagated to them at this stage. Apologies for not including these details, we will clarify them in the final manuscript.

---

> > ### Comment · Reviewer_fqdv · 2025-08-05
> >
> > I thank the authors for their detailed answer and have no further questions open for discussion. I remain positive about the paper and its contributions.

---

### Official Review · Reviewer_AwCB · 2025-07-05

**Clarity:** 3
**Significance:** 3
**Originality:** 3
**Rating:** 3
**Confidence:** 4

**Summary:**

The authors proposed a method for semantic correspondence lifts sparse 2D keypoints into a shared canonical 3D space using monocular depth estimation. By learning a continuous, category-specific canonical manifold, the method aims to capture object geometry and guide dense correspondence learning without requiring explicit 3D supervision or camera pose information. Alongside the method, the authors introduce SPair-U, an extension of the SPair-71k benchmark, which includes additional keypoints not used during training. This allows for explicit evaluation of generalization to unseen points. Experiments show that the proposed method outperforms supervised baselines on these unseen keypoints, demonstrating improved generalization.

**Questions:**

Please see the weaknesses section above.

**Ethical Concerns:**

["NO or VERY MINOR ethics concerns only"]

**Limitations:**

Yes

**Quality:**

3

**Strengths And Weaknesses:**

Strengths
- Shifting the focus from sparse, supervised alignment to dense, geometry-aware representation learning
- Leveraging off-the-shelf components like monocular depth estimation (Moge) and segmentation masks (SAM), and is integrated into an existing model (Geo-SC) in a modular way
- The introduction of the SPair-U benchmark;  exposing the overfitting tendency of standard SC models and enabling evaluation on unseen keypoints

Weaknesses
- The method introduces several additional sources of supervision (depth maps, segmentation masks, canonical 3D supervision) that are not available to baseline SC models, raising concerns about fairness in the performance comparison. The use of off-the-shelf modules like Moge and SAM means the observed gains could stem as much from auxiliary data as from the core method itself. This issue is not adequately discussed, and no control experiments are provided to isolate the contribution of each added component.
- As for the benchmark, while SPair-U introduces new test keypoints, the extension is somewhat limited: only a few extra keypoints are added only for testing data. For the practical utility of the benchmark as a community resource,  it is desirable to provide dense annotations across all images, including those in the training set, by leveraging oracle-like 3D canonical representations.

---

> ### Author Rebuttal · Authors · 2025-07-30
>
> We thank **AwCB** for their insightful review and for highlighting our shift toward geometry-aware learning and the value of SPair-U for revealing overfitting. As an initial remark, we would like to clarify that our paper does more than just introducing a new semantic correspondence approach and new benchmark dataset. Our goal is to expose a fundamental flaw in the way supervised semantic correspondence pipelines are trained and evaluated, i.e. **issues in the general task design** are the reason behind the generalization gap observed in models. As a result, GeoSC - the top performing model on SPair-71k at the time we wrote our paper - trains a supervised feature predictor on top of frozen DINOv2 and SD features. However, when evaluated on SPair-U, it performs *worse* than simply using the DINOv2 and SD features alone (see Tables 1 and 2). This indicates that **none** of the knowledge it has acquired transfers to new semantics, meaning it fails to  solve anything beyond sparse keypoint detection. We believe this finding is a key contribution of our work, with significant implications for future model and benchmark design.
>
> **[AwCB-1] Extra segmentation and depth supervision.**
> We do acknowledge this additional supervision in our limitations section (see L308). Using the depth and segmentation predictions at training time from pre-trained models to build 3D-aware representations is “free” supervision and one or the other has been adopted before in related works [3, 27, 42, 46]. In particular, concurrent work [46] uses both, even requiring test-time segmentation masks, while in our case they are only used during training. While this regime could be considered unfair, our central claim is that the prevailing evaluation setup is itself biased: top-performing models on SPair-71k fail on SPair-U (Fig. 1). In that context, the purpose of our model is to demonstrate that using a 3D-informed representation can help bridge the generalization gap.
>
> As stated in our supplementary PDF (L24), it is challenging to isolate the contributions of segmentation (SAM) and depth (MoGE) within our pipeline, as both are required to compute our geometric losses. Removing either of them would prevent the computation of Eq. 6 and revert our model back to Geo-SC. Conversely, as previous approaches generally rely on sparse 2D-based models, there is no straightforward way to directly incorporate depth or segmentation in their design.  Nonetheless, as requested, we performed an additional experiment to study the contribution of adding depth and segmentation by comparing our full model with Geo-SC (Zhang et al. CVPR 2024), as well as by ablating the Geo-SC-specific losses from our model, and comparing it to a simple supervised approach:
>
> | PCK@0.1$_{bbox}$ | SPair-71k| SPair-U|
> |-|-:|-:|
> |DINOv2+SD (unsupervised, for reference)| 61.1 | 59.4|
> |DINOv2+SD| 76.5 | 60.0|
> |DINOv2+SD+Geo-SC (i.e., Geo-SC)| 85.6| 57.1|
> |DINOv2+SD+canonical prototype| 75.9 | 66.0 |
> DINOv2+SD+Geo-SC losses+canonical prototype (i.e., Ours)| 85.4| 66.1|
>
> The results show a clear pattern, i.e., adding our canonical prototype loss results in a very small drop in performance on seen keypoints (i.e., SPair-71k), which we attribute to the models having less capacity to fully overfit the training keypoint supervision, but endows them with the ability to generalize much better to unseen points (i.e., SPair-U). Conversely, the Geo-SC losses allow models to perform really well on seen keypoints but its effect on generalization ranges from harmful to null (see “DINOv2+SD” vs. “DINOv2+SD+Geo-SC”). These results also demonstrate that our contributions do not require Geo-SC to work, as they also boost performance of the supervised baseline on unseen keypoints (see “DINOv2+SD” vs. “DINOv2+SD+canonical prototype”). We will add this ablation to the revised text. Thanks for the suggestion.
>
>
> **[AwCB-2] SPair-U limitations.**
> We acknowledge the potential limitation regarding dataset size on L300. The existing SPair-71k dataset contains a total of 88,328 test keypoint pairs and our new SPair-U dataset contains 19,990, which is still significant. SPair-U is an evaluation tool to expose the lack of generalization of sparse supervised correspondence models, which we believe it does effectively. Importantly, it is not for training models, as making these points available during training would prevent assessing generalization. A particular advantage of SPair-U is that it can be used to easily test any model that supports evaluation on SPair-71k, as it simply requires swapping the annotation files, and provides a clear visualization of the gap, e.g., “seen” versus “unseen” points performance.
>
> We fully agree that a dense labeling with 3D canonical representation would be valuable for the community in the long run. However, this is not a trivial task, and is in fact an actively studied research problem. Collecting “oracle” dense annotations is extremely labour intensive and limited to few real object categories as it typically relies on using 3D meshes (see related works [21,31]).

---

> > ### Comment · Reviewer_AwCB · 2025-08-06
> >
> > While I appreciate the authors’ rebuttal and the clarification of their primary goal—namely, to demonstrate that incorporating a 3D-informed representation (via SAM + depth) can help bridge the generalization gap—I remain unconvinced that the current version of the paper meets the bar for publication.
> >
> > First, the performance gains attributed to the proposed method still appear to stem largely from the addition of depth cues and SAM masks. These components are known to provide strong priors, and their effectiveness is not surprising. However, the paper does not convincingly isolate how much of the improvement comes from the 3D-aware design itself versus these auxiliary components. The lack of an ablation study or any form of quantitative disentanglement makes it difficult to assess the core methodological contribution fairly.
> >
> > Second, the comparisons to prior work remain insufficient. Although the authors state that their method is “training-free,” many of the baselines they compare against do not operate under the same constraints. This makes it challenging to evaluate the benefit of the proposed design on equal footing. Moreover, there are several recent works that attempt to incorporate geometry or depth in segmentation or detection tasks, and I did not see a sufficiently broad discussion or positioning of this work in that context.
> >
> > While I understand that the authors may still be working on a more thorough component-wise analysis, the current submission does not yet provide enough evidence to support its core claims. For instance, without a quantitative justification of each module’s role (e.g., depth encoder, graph structure, clustering strategy), it is hard to determine whether the proposed framework offers insights beyond an intuitive stacking of known effective tools.

---

> > > ### Author Response · Authors · 2025-08-06
> > >
> > > Thank you for the additional remarks.
> > >
> > > **Central contribution**
> > > We would like to re-state that our primary goal is exposing the generalization gap present in recent approaches, and showing that it stems from a flaw in the task design in order to help the design of future models and benchmarks. Regarding the comment that “the current submission does not yet provide enough evidence to support its **core claims**”, we believe that our evaluation of these points is thorough and comprehensive. However, if there are specific missing experiments that would improve our paper, we would be happy to add them.
> > >
> > > **Ablations of segmentation and depth**
> > > MoGE and SAM are only used offline to obtain depth and mask predictions, we do not use their intermediate features, so their learned priors are not added to our model. As we noted in our rebuttal, this is “free supervision” in the sense that we do not require any additional manual supervision, instead use the predictions from these pre-trained models. Similar supervision has been used in existing work (see response [AwCB-1]).
> > > In the ablation study provided in the rebuttal (see response [AwCB-1]), adding our canonical prototype raises PCK@0.1 on SPair-U from 60.0 to 66.0 while leaving performance on seen keypoints mostly unchanged (75.9 versus 76.5). This shows that the contribution of the added supervision is isolated to new semantics only.
> > >
> > > **Related works incorporating geometry or depth**
> > > In our related works section, we discuss a method that uses depth ([3]), two that both use segmentation ([27,46]), and one that uses both, including during inference ([42]). If there are specific works we have missed, please do not hesitate to name them and we would be happy to add and discuss them.
> > >
> > > **Components of our method**
> > > Perhaps there was some misunderstanding as our method does *not* employ a “graph structure” or “clustering strategy”. In addition, we only use MoGE to extract depth maps as a pre-processing step, therefore our model does not make use of a “depth encoder”. Finally, our method does not claim to be “training-free”. On the contrary, we design an additional geometric loss (Eq. 6) that trains the feature predictor $\phi$ alongside the typical semantic correspondence losses. If the text of the paper is unclear on this point, please do not hesitate to point out the specific passages in question and we can update them for the final revised version.

---

### Decision · Program_Chairs · 2025-09-17

**Decision:**

Accept (poster)

**Comment:**

This paper investigates the generalization ability of semantic correspondence methods beyond sparsely annotated training keypoints and proposes a novel approach for learning dense correspondences by lifting 2D keypoints into a canonical 3D space using monocular depth estimation. Additionally, the authors introduce a new benchmark, SPair-U, with novel keypoint annotations to better assess the issue of generalization. There were several concerns regarding motivation, assumptions, analyses, and comparisons, and there were a rebuttal and follow-up discussion. After post-rebuttal discussions, the paper received mixed reviews, one accept, two borderline accepts, and two borderline rejects. The main remaining concerns were the lack of in-depth analyses/comparison (reviewer AwCB) and the implicit assumption of the same shape for the same category (reviewer UJnP). After carefully reading the review, the rebuttal, the discussions, and the paper, AC finds that while the reviewers’ points are valid and reasonable, the authors’ rebuttal addresses them relatively well, and the remaining issues are not critical enough to warrant rejection. For example, the global rigidity assumption is fairly strong for non-rigid objects indeed, but not a completely unreasonable assumption, and the authors demonstrated well that it helps close the generalization gap of existing supervised semantic correspondence methods; the effect of the implicit assumption may motivate further development into a more relaxed assumption of shapes in future work. AC also feels positive about the impact and contributions of the work, which reveal and investigate the interesting issue of existing work on the important vision task. Therefore, AC recommends accepting this work. AC strongly encourages the authors to incorporate the reviewers’ constructive comments and discussions into their final manuscript to clarify their contributions and include missing related work, relevant analyses, and discussions.